# De-randomizing MCMC dynamics with the diffusion Stein operator

**Zheyang Shen**[1]    **Markus Heinonen**[1]    **Samuel Kaski**[1,2]

[1]Helsinki Institute for Information Technology, HIIT,
Department of Computer Science, Aalto University, Finland
[2]Department of Computer Science, University of Manchester
{zheyang.shen, markus.o.heinonen, samuel.kaski}@aalto.fi

## Abstract

Approximate Bayesian inference estimates descriptors of an intractable target distribution – in essence, an optimization problem within a family of distributions. For example, Langevin dynamics (LD) extracts asymptotically exact samples from a diffusion process because the time evolution of its marginal distributions constitutes a curve that minimizes the KL-divergence via steepest descent in the Wasserstein space. Parallel to LD, Stein variational gradient descent (SVGD) similarly minimizes the KL , albeit endowed with a novel Stein-Wasserstein distance, by *deterministically* transporting a set of particle samples, thus de-randomizes the stochastic diffusion process. We propose de-randomized kernel-based particle samplers to all diffusion-based samplers known as MCMC dynamics. Following previous work in interpreting MCMC dynamics, we equip the Stein-Wasserstein metric with a fiber-Riemannian Poisson structure, with the capacity of characterizing a fiber-gradient Hamiltonian flow that simulates MCMC dynamics. Such dynamics discretize into generalized SVGD (GSVGD), a Stein-type deterministic particle sampler, with particle updates coinciding with applying the *diffusion Stein operator* to a kernel function. We demonstrate empirically that GSVGD can de-randomize complicated MCMC dynamics, which combine the advantages of auxiliary momentum variables and Riemannian structure, while maintaining the high sample quality from an interacting particle system.

## 1   Introduction

Evaluating an un-normalized target distribution $\pi$ is a centerpiece of Bayesian inference, due to its ubiquitous presence in posterior distributions. Markov chain Monte Carlo (MCMC) methods fulfill this objective by generating asymptotically exact random samples from the distribution, a significant subset of which involves discretization of continuous-time diffusion processes, most notably Langevin diffusion, stochastic gradient Hamiltonian Monte Carlo (HMC) (Chen et al., 2014) and their further generalizations, which we collectively call MCMC *dynamics* (Ma et al., 2015). Despite its simplicity and theoretical soundness, this diffusion-based sampling often suffers from slow convergence and small effective sample sizes, largely due to the auto-correlation of the samples.

As an alternative to the simulation of stochastic systems, particle-based variational inference (Liu and Wang, 2016; Chen et al., 2018; Liu et al., 2019a) partially addresses the shortcomings of MCMC by replacing the Langevin diffusion, the simplest MCMC dynamics, with a deterministic interacting particle system that transports a set of interacting particles towards the target distribution. Theoretically speaking, Langevin diffusion encodes an evolution of density that minimizes the KL-divergence through steepest descent in the 2-Wasserstein space (Jordan et al., 1998), and particle-

35th Conference on Neural Information Processing Systems (NeurIPS 2021).

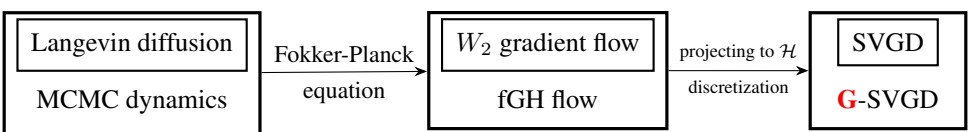

Figure 1: A diagram showing the contribution of our paper: we extend and generalize previous work linking Langevin diffusion to gradient flow on the 2-Wasserstein metric $W_2$ through the Fokker-Planck equation, and the $W_2$ gradient flow linking to SVGD (Liu and Wang, 2016) through projection onto an RKHS; MCMC dynamics (Ma et al., 2015) are interpreted as a fiber-gradient Hamiltonian (fGH) flow on $W_2$ (Liu et al., 2019b), and a projection onto an RKHS yields GSVGD.

based variational inference (PARVI) approximates such evolution using reproducing kernel Hilbert space (RKHS) (Liu, 2017; Liu et al., 2019a), thus de-randomizing the Langevin diffusion process.

Given the elegant theoretical link between Langevin diffusion and PARVI, one crucial question arises: can we leverage the advantages of general MCMC dynamics onto de-randomized particle systems? While it is difficult to formulate them as a direct Wasserstein gradient flow, Liu et al. (2019b) propose an interpretation for "regular" MCMC dynamics that augments the 2-Wasserstein space with a fiber bundle, forming a fiber-Riemannian Poisson manifold, under which MCMC dynamics follows a Hamiltonian flow on the fiber bundle, and a gradient flow on each fiber.

We show that adapting the fiber-Riemannian Hamiltonian flow to the Stein-Wasserstein metric (Liu, 2017; Duncan et al., 2019) yields a vector field in the form of applying the *diffusion Stein operator* to the kernel function $k(\cdot, \boldsymbol{\theta})$. A discretization of this vector field gives generalized SVGD (GSVGD), an interacting particle system that generalizes SVGD, with particle updates balancing an attractive force maximizing the log-likelihood and a repulsive force preventing a "mode collapse" of particles. We further demonstrate that the connection drawn between LD and SVGD (Liu and Wang, 2016; Liu, 2017; Liu et al., 2019a) is retraced by GSVGD, reaffirming our claim that GSVGD mirrors SVGD in approximating a larger class of MCMC diffusion processes. Within the generality of our framework, we can develop PARVI algorithms that exploit two key types of possible acceleration in MCMC dynamics: auxiliary momentum variables and an adaptive Riemannian parametrization that allows for fast and efficient exploration of the probability space (Girolami and Calderhead, 2011), as shown in Table 1.

| MCMC dynamics | $\mathbf{A}$ | $\mathbf{C}$ | auxiliary variable | Riemannian | PARVI variant |
|---|---|---|---|---|---|
| SGLD (Welling and Teh, 2011) | $\mathbf{I}$ | $\mathbf{0}$ | ✗ | ✗ | SVGD (Liu and Wang, 2016) Blob (Chen et al., 2018) |
| SGRLD (Girolami and Calderhead, 2011) | $\mathbf{G}(\boldsymbol{\theta})^{-1}$ | $\mathbf{0}$ | ✗ | ✓ | Riemannian SVGD (Liu and Zhu, 2017) |
| SGHMC (Chen et al., 2014) | $\begin{pmatrix} \mathbf{0} & \mathbf{0} \\ \mathbf{0} & A\mathbf{I} \end{pmatrix}$ | $\begin{pmatrix} \mathbf{0} & -\mathbf{I} \\ \mathbf{I} & \mathbf{0} \end{pmatrix}$ | ✓ | ✗ | HMC-blob (Liu et al., 2019b) |
| SGRHMC (Ma et al., 2015) | $\begin{pmatrix} \mathbf{0} & \mathbf{0} \\ \mathbf{0} & \mathbf{G}(\boldsymbol{\theta})^{-1} \end{pmatrix}$ | $\begin{pmatrix} \mathbf{0} & -\mathbf{G}(\boldsymbol{\theta})^{-1/2} \\ \mathbf{G}(\boldsymbol{\theta})^{-1/2} & \mathbf{0} \end{pmatrix}$ | ✓ | ✓ | this work (SGRHMC-Stein) |

| (a) LD path | (b) SVGD | (c) SGHMC path | (d) SGHMC Stein |
|---|---|---|---|

Table 1: An overview of MCMC dynamics along with their PARVI approximations. Between **(a), (c)** and **(b), (d)**, MCMC dynamics simulate diffusion processes (shown by one instance of a particle trajectory) that *temporally* draw samples from the target distribution (samples marked by scatterplots); PARVI approximates the density evolution of MCMC diffusions with a set of particles with deterministic interactions; Between **(a), (b)** and **(c), (d)**, the two systems, namely LD (Welling and Teh, 2011) and SGHMC (Chen et al., 2014), follow different density evolutions, as **(a), (b)** describe a gradient flow and **(c), (d)** describe a fiber-gradient Hamiltonian flow.

# 2 MCMC dynamics – the relevant bits

In this paper, we consider the problem of extracting samples $\theta \in \mathbb{R}^D = \Omega$ from an un-normalized target distribution $\pi(\theta) \propto e^{-H(\theta)}$. We begin by reviewing the key characteristics of MCMC dynamics, with particular emphasis on Langevin dynamics and its Fokker-Planck equation (FPE), the partial differential equation (PDE) that depicts the time evolution of a diffusion process. As the FPE of LD conforms to a gradient flow structure in the 2-Wasserstein metric space of probability measures $(\mathcal{P}(\Omega), W_2)$ (Jordan et al., 1998), we then briefly cover gradient flow on $\mathcal{P}(\Omega)$ as an infinite-dimensional Riemannian manifold. Through the lens of gradient flow, we can see SVGD as a deterministic interacting particle system that approximates the gradient flow of LD through (i) gradient flow on a novel Stein-Wasserstein metric (Liu, 2017) or (ii) projecting the gradient flow direction onto an RKHS (Liu et al., 2019a). With Wasserstein gradient flow (WGF) neatly linking to both LD and SVGD, we see that a diffusion process LD and an interacting particle system SVGD are, respectively, a stochastic instance and a deterministic approximation of the same Wasserstein gradient flow.

**Notations**: We use $\nabla f$ to denote the gradient of a scalar-valued function, and $\nabla \cdot \mathbf{f}$ the divergence of a vector-valued function, $\nabla \cdot \mathbf{A}$ applies the divergence operator to each row of a matrix-valued function, $\dot{\rho}_t$ the "partial derivative" with respect to $t$.

## 2.1 Langevin dynamics and its Fokker-Planck equation

In this paper, we consider MCMC dynamics in the form of Itō diffusion processes following the stochastic differential equation (SDE) formula

$$d\theta_t = \mathbf{f}(\theta_t)dt + \sqrt{2\Sigma(\theta_t)}d\mathbf{W}_t, \tag{1}$$

consisting of drift coefficient $\mathbf{f} : \mathbb{R}^D \mapsto \mathbb{R}^D$, diffusion coefficient $\sqrt{2\Sigma}$ and a $D$-dimensional Brownian motion $\mathbf{W}_t$. Ma et al. (2015) provide a complete recipe of all Itō diffusion processes converging to the target measure $\pi$. The simplest MCMC dynamics takes the form of Langevin diffusion (Langevin, 1908), which moves towards higher densities with $\mathbf{f}(\theta) = \nabla \log \pi(\theta)$ perturbed by white noise $\Sigma = \mathbf{I}$. Given an initial distribution $\theta_0 \sim \rho_0$, the Fokker-Planck equation describes the time evolution of the density of $\theta_t$ (Risken, 1996)

$$\dot{\rho}_t + \nabla \cdot (\rho_t \nabla \log \pi) - (\nabla \nabla) : (\rho_t \mathbf{I}) = 0, \tag{2}$$

where $\mathbf{X} : \mathbf{Y} = \text{tr}(\mathbf{X}^\top \mathbf{Y})$. Given that $(\nabla \nabla) : \mathbf{M} = \nabla \cdot (\nabla \cdot \mathbf{M})$, we can rewrite (2) as

$$\dot{\rho}_t = \nabla \cdot (\nabla \rho_t - \rho_t \nabla \log \pi) = \nabla \cdot \left( \rho_t \nabla \underbrace{\frac{\delta \text{KL}\left[\rho_t \| \pi\right]}{\delta \rho_t}}_{\text{first variation of KL}\left[\rho_t \| \pi\right]} \right), \tag{3}$$

a crucial step in developing the Wasserstein gradient flow perspective of LD, as $\nabla \delta E / \delta \rho$ coincides with the differential of functionals induced by the Wasserstein metric.

## 2.2 Gradient flow on $(\mathcal{P}(\Omega), W_2)$

The conventional gradient flow (or the steepest descent curve) that minimizes a smooth function $F : \mathbb{R}^D \mapsto \mathbb{R}$ follows the PDE: $\dot{\mathbf{x}}_t + \nabla F(\mathbf{x}_t) = 0$. Defining gradient flow on $\mathcal{P}(\Omega)$, informally written as $\dot{\rho}_t + \nabla_W E(\rho_t) = 0$, requires a definition of the differentiation $\nabla_W$, which requires an understanding of the Wasserstein metric, i.e., the inner product $g_\rho$ defined on its tangent space.

To simplify the discussion, we restrict the discussion on measures with a density function and with finite second-order moments. For each $\rho$, the tangent space constitutes smooth functions integrating to zero, $T_\rho \mathcal{P}(\Omega) = \{f | f \in C^\infty(\Omega), \int f(\theta)d\theta = 0\}$, in that a curve on $\mathcal{P}(\Omega)$ preserves volume. The cotangent space constitutes an equivalence class of smooth functions in differing constant, noted as the quotient space $T_\rho^* \mathcal{P}(\Omega) = C^\infty(\Omega)/\mathbb{R}$. We characterize the inner product space using the metric tensor $G(\rho) : T_\rho \mathcal{P} \mapsto T_\rho^* \mathcal{P}$, a one-to-one mapping between elements of the tangent space and those of the cotangent space. The inner product $g_\rho$ is defined by the inverse of the metric tensor, often denoted as Onsager operators (Onsager, 1931)

$$g_\rho(f_1, f_2) = \int f_1 G(\rho) f_2 d\theta = \int \phi_1 G(\rho)^{-1} \phi_2 d\mathbf{x}, \quad \phi_1 = G(\rho) f_1, \ \phi_2 = G(\rho) f_2. \tag{4}$$

The Wasserstein Onsager operator takes the form $G(\rho)^{-1} : \phi \mapsto -\nabla \cdot (\rho \nabla \phi)$. In the context of Wasserstein gradient flow, we define $\nabla_W E(\rho)$ as $G(\rho)^{-1} \frac{\delta E(\rho)}{\delta \rho}$, leading to the formulation:

$$0 = \dot{\rho}_t - G(\rho)^{-1} \frac{\delta E(\rho_t)}{\delta \rho_t} = \dot{\rho}_t + \nabla \cdot \left( \rho_t \nabla \frac{\delta E(\rho_t)}{\delta \rho_t} \right). \tag{5}$$

The first variation of KL $[\rho \parallel \pi]$ takes the form of $\frac{\delta \text{KL}[\rho \parallel \pi]}{\delta \rho} = \log \rho / \pi + 1$, leading to the conclusion by Jordan et al. (1998) that the time evolution of the Langevin diffusion follows a curve of steepest descent in the Wasserstein space, laying the foundation for approximation using particle interaction.

## 2.3 SVGD as gradient flow

PARVI transports a set $N$ of interacting particles $\left\{ \boldsymbol{\theta}_t^{(i)} \right\}_{1 \leq i \leq N}$ towards the target distribution over time. Take SVGD for example, denoting the empirical measure at time $t$ as $\hat{\rho}_t = \frac{1}{N} \sum_{i=1}^N \delta_{\boldsymbol{\theta}_t^{(i)}}$, SVGD (Liu and Wang, 2016) follows the update rule

$$\dot{\boldsymbol{\theta}}_t = \mathbb{E}_{\boldsymbol{\theta}' \sim \hat{\rho}_t} \left[ k(\boldsymbol{\theta}_t, \boldsymbol{\theta}') \nabla \log \pi(\boldsymbol{\theta}') + \nabla_2 k(\boldsymbol{\theta}_t, \boldsymbol{\theta}') \right] = \mathbf{v}_{\mathcal{H}}(\boldsymbol{\theta}_t | \hat{\rho}_t), \tag{6}$$

where $k(\cdot, \cdot)$ defines a positive-definite kernel, and $\nabla_2$ denotes the gradient of the second argument. The FPE of SVGD can be written as

$$\dot{\rho}_t - \nabla \cdot \left( \rho_t \mathcal{K}_\rho \nabla \frac{\delta \text{KL}[\rho_t \parallel \pi]}{\delta \rho_t} \right) = 0, \tag{7}$$

where $\mathcal{K}_\rho$ is an integral operator: $\mathcal{K}_\rho \mathbf{f}(\boldsymbol{\theta}) = \int k(\boldsymbol{\theta}, \boldsymbol{\theta}') \mathbf{f}(\boldsymbol{\theta}') \mathrm{d}\rho(\boldsymbol{\theta})$. We shall denote the RKHS defined by $k$ as $\mathcal{H}$. While significantly different from LD at first glance, SVGD approximates the WGF of LD by

- kernelizing the Wasserstein Onsager operator to give $G_{\mathcal{H}}(\rho)^{-1} : \phi \mapsto -\nabla \cdot (\rho \mathcal{K}_\rho \nabla \phi)$, thus forming the Stein-Wasserstein metric $W_{\mathcal{H}}$ (Liu, 2017; Duncan et al., 2019);

- projecting the gradient flow vector field $\mathbf{v}(\rho) = -\nabla \frac{\delta \text{KL}[\rho \parallel \pi]}{\delta \rho} = \nabla \log \pi - \nabla \log \rho$ onto $\mathcal{H}$ (Liu et al., 2019a).

The interpretation of SVGD yields valuable insights. While gradient flow (7) on $(\mathcal{P}(\Omega), W_2)$ yields no closed-form energy functional (Chen et al., 2018), it behooves to absorb the operator $\mathcal{K}_\rho$ into the definition of the Stein-Wasserstein metric to guarantee a tractable energy functional. In the meantime, the kernelization trick transforms a gradient flow (2) simulated by a diffusion process into an approximate deterministic transportation of particles – in other words, de-randomizes it.

## 2.4 MCMC dynamics as fiber-gradient Hamiltonian flow

The concept of *flow* on $(\mathcal{P}(\Omega), W_2)$ is a generalization of the gradient flow: given a vector field $\mathbf{v} : \mathcal{P}(\Omega) \mapsto \bigcup_{\rho \in \mathcal{P}} T_\rho$, the flow of $\mathbf{v}$ is defined as $\dot{\rho}_t = \mathbf{v}(\rho_t)$. Liu et al. (2019b) interpret "regular" MCMC dynamics on $(\mathcal{P}(\Omega), W_2)$ as a flow, by combining gradient flow and Hamiltonian flow on the Wasserstein space, yielding a fiber-Riemannian manifold structure of $\mathcal{P}(\Omega)$, a fiber bundle consisting of Riemannian manifolds. In the context of a "regular" MCMC dynamics taking the form of underdamped Langevin dynamics (Chen et al., 2014), the diffusion matrix $\mathbf{A}$ determines the gradient flow on each fiber, and the curl matrix $\mathbf{C}$ determines the Hamiltonian flow on the fiber bundle. The Hamiltonian flow keeps KL $[\rho \parallel \pi]$ constant while encouraging fast exploration of the probability space; the fiber gradient flow minimizes the KL. The fiber-gradient Hamiltonian flow determines a PARVI with the particle update

$$\dot{\boldsymbol{\theta}}_t = (\mathbf{A}(\boldsymbol{\theta}_t) + \mathbf{C}(\boldsymbol{\theta}_t)) \left( \nabla \log \pi(\boldsymbol{\theta}_t) - \widehat{\nabla} \log \rho_t(\boldsymbol{\theta}_t) \right) = \mathbf{v}^{\mathbf{A}, \mathbf{C}}(\rho_t), \tag{8}$$

where the intractable $\widehat{\nabla} \log \rho_t$ is approximated via the Blob method (Carrillo et al., 2019).

## 2.5 Stein's method and other relevant works

Our work enriches the practical application of Stein's method (Stein, 1972), which studies the class of operators that maps functions to ones with expectation zero w.r.t. a distribution $\pi$. Gorham et al. (2019) draw from the findings of the generator method (Barbour, 1988, 1990; Gotze, 1991) and note the same property with infinitesimal generators of Feller processes and their stationary measures, which further extends into a mapping between diffusion-based MCMC sampling and Stein operators, noted as *diffusion Stein operators*. In this work, we extend beyond sample quality measurement (Gorham et al., 2019) and parameter estimation (Barp et al., 2019) and establish an application of the diffusion Stein operator as a deterministic alternative of MCMC dynamics.

Myriad works (Chen et al., 2018; Liu et al., 2019a; Zhang et al., 2020) seek alternatives to approximating the gradient flow (2) through kernelization. Notably, Chen et al. (2018); Liu et al. (2019a) construct PARVI algorithms by approximating the term $\nabla \log \rho_t(\boldsymbol{\theta}_t)$ in the gradient flow, namely with Blob method (Carrillo et al., 2019), kernel density estimation (Liu et al., 2019a) and Stein gradient estimator (Li and Turner, 2017). The de-randomization of underdamped LD connects to the viewpoint of accelerating PARVI methods. Ma et al. (2019) demonstrate that underdamped LD (Chen et al., 2014) accelerates the steepest descent steps taken by the overdamped LD, forming an analog of Nesterov acceleration for MCMC methods. Wang and Li (2019) present a framework for Nesterov's accelerated gradient method in the Wasserstein space, which consists of augmenting the energy functional with the kinetic energy of an additional momentum variable.

## 3 PARVI for MCMC dynamics – a general recipe

In this section, we outline the main contribution of this work, which traces the roadmap delineated by previous works to explore the flow interpretation, as well as the approximation by interacting particle systems of general form MCMC dynamics (Ma et al., 2015) in the form of Itō diffusion:

$$\mathbf{f}(\boldsymbol{\theta}) = \frac{1}{\pi(\boldsymbol{\theta})} \nabla \cdot \left[ \pi(\boldsymbol{\theta}) \left( \mathbf{A}(\boldsymbol{\theta}) + \mathbf{C}(\boldsymbol{\theta}) \right) \right], \tag{9}$$

$$\boldsymbol{\Sigma}(\boldsymbol{\theta}) = \mathbf{A}(\boldsymbol{\theta}), \tag{10}$$

where $\mathbf{A}$ and $\mathbf{C}$ are positive-semidefinite and skew-symmetric matrix-valued functions, respectively. This general framework covers *all* continuous Itō diffusion processes with stationary distribution $\pi$, most notably LD, stochastic gradient HMC (Chen et al., 2014), stochastic gradient Nosé-Hoover thermostat (SGNHT) (Ding et al., 2014) and stochastic gradient Riemannian Hamiltonian Monte Carlo (SGRHMC) (Ma et al., 2015).

We demonstrate in this section that applying the fiber-gradient Hamiltonian flow structure of MCMC dynamics to the Stein-Wasserstein metric yields a flow on $(\mathcal{P}(\Omega), W_{\mathcal{H}})$ that discretizes into GSVGD, a PARVI that updates particles with the diffusion Stein operator (Gorham et al., 2019), suggesting that the infinitesimal generator of MCMC diffusion processes offers a "kernel smoothing" de-randomization. Apart from the Stein-Wasserstein metric, the analog of GSVGD generalizing SVGD extends to other interpretations of SVGD, namely that GSVGD takes steepest descent minimizing the KL-divergence in incremental transformation of particles, and that that GSVGD projects the fiber-gradient Hamiltonian flow onto an RKHS.

### 3.1 Fiber-gradient Hamiltonian flow on $(\mathcal{P}(\Omega), W_{\mathcal{H}})$

Similar to (2), we derive the FPE of MCMC dynamics (9)-(10)

$$\dot{\rho}_t = -\nabla \cdot (\rho_t \mathbf{f}) + (\nabla\nabla) : (\rho_t \mathbf{A}) \tag{11}$$

$$= -\nabla \cdot (\rho_t \mathbf{f}) + (\nabla\nabla) : (\rho_t \mathbf{A}) + \underbrace{(\nabla\nabla) : (\rho_t \mathbf{C})}_{=0} \tag{12}$$

$$= \nabla \cdot \left( \rho_t (\mathbf{A} + \mathbf{C}) \nabla \log \frac{\rho_t}{\pi} \right) = \nabla \cdot \left( \rho_t (\mathbf{A} + \mathbf{C}) \nabla \frac{\delta \mathrm{KL}\left[\rho_t \parallel \pi\right]}{\delta \rho_t} \right), \tag{13}$$

which is a curve in $\mathcal{P}(\Omega)$ following the continuity equation $\dot{\rho}_t + \nabla \cdot (\rho_t \mathbf{v}_t) = 0, \mathbf{v}_t = (\mathbf{A} + \mathbf{C}) \nabla \log \rho_t / \pi$.

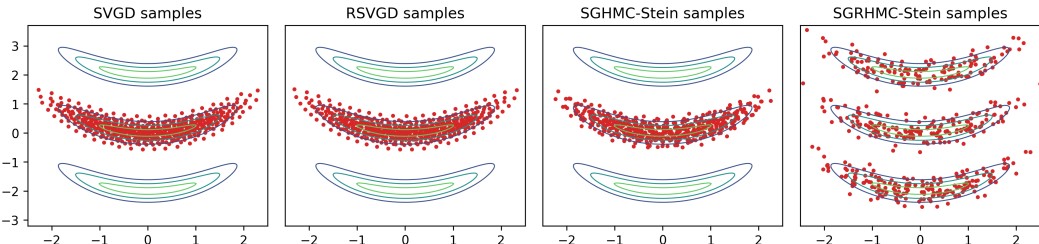

Figure 2: Particle samples of an underlying 2-dimensional correlated mixture $\pi$ extracted by SVGD (Liu and Wang, 2016), Riemannian SVGD (Liu and Zhu, 2017), SGHMC-Stein and SGRHMC-Stein. While all methods explore the mode nearest to initialization, only RHMC explores all three modes.

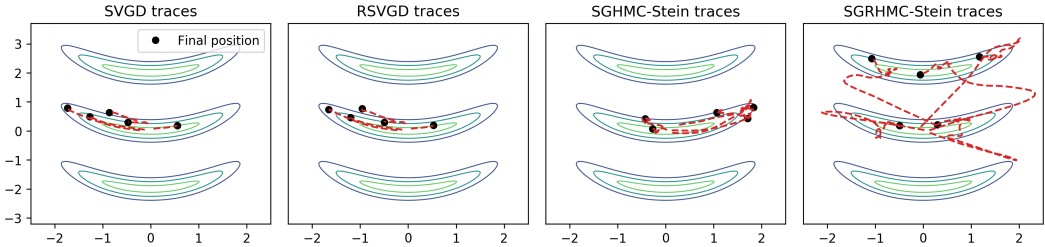

Figure 3: A visualisation of sampler trajectories on the tri-Gaussian density $\pi$. With the help of momentum variable and efficient Riemannian parametrization of $(\mathbf{A}, \mathbf{C})$, SGRHMC-Stein aggressively explores the space, while other methods cannot escape the local component.

We can derive GSVGD from the fiber-Riemannian manifold perspective of $(\mathcal{P}(\Omega), W_{\mathcal{H}})$ (Liu et al., 2019b). The Riemannian structure of the Stein-Wasserstein metric induces the tangent space $T_\rho \mathcal{P}(\Omega) = \overline{\{\mathcal{K}_\rho \nabla f \mid f \in C^\infty(\Omega)\}}^{\mathcal{H}^D}$. An orthogonal projection $\mathfrak{p}_\rho$ from $\mathcal{L}^2_\rho(\Omega)$ to $T_\rho \mathcal{P}$ is uniquely defined as such vector in $T_\rho \mathcal{P}$ that satisfies $\nabla \cdot (\rho \mathcal{K}_\rho \phi) = \nabla \cdot (\rho \mathfrak{p}_\rho(\phi))$. Using Theorem 5 from Liu et al. (2019b), we arrive at a fiber-gradient Hamiltonian flow

$$\mathcal{W}_{\mathrm{KL}_\pi}(\rho) = \mathfrak{p}_\rho \left( (\mathbf{A} + \mathbf{C}) \nabla \log \pi / \rho \right). \tag{14}$$

It is straightforward to verify that $\mathbf{v}_{\mathcal{H}}^{\mathbf{A},\mathbf{C}}(\rho) = \mathcal{K}_\rho (\mathbf{A} + \mathbf{C}) \nabla \log \pi / \rho$ induces the same evolution of distribution as $\mathcal{W}_{\mathrm{KL}_\pi}(\rho)$.

### 3.2  fRH flow induces the diffusion Stein operator

The Wasserstein flow on $(\mathcal{P}(\Omega), W_{\mathcal{H}})$ following $\dot{\rho}_t = \nabla \cdot \left( \rho_t \mathbf{v}_{\mathcal{H}}^{\mathbf{A},\mathbf{C}}(\rho_t) \right)$ requires spatial and temporal discretization suitable for a particle update. While $\mathbf{v}_{\mathcal{H}}^{\mathbf{A},\mathbf{C}}$ constitutes a vector field on $\mathcal{P}(\Omega)$, its calculation does not involve an explicit term of $\nabla \log \rho$ inside the expectation.

$$\mathbf{v}_{\mathcal{H}}^{\mathbf{A},\mathbf{C}}(\rho) = \mathcal{K}_\rho \left[ (\mathbf{A} + \mathbf{C}) \nabla \log \pi / \rho \right] \tag{15}$$

$$= \mathbb{E}_{\boldsymbol{\theta}' \sim \rho} k(\cdot, \boldsymbol{\theta}')(\mathbf{A}(\boldsymbol{\theta}') + \mathbf{C}(\boldsymbol{\theta}')) \nabla \log \pi(\boldsymbol{\theta}') - \int k(\cdot, \boldsymbol{\theta}')(\mathbf{A}(\boldsymbol{\theta}') + \mathbf{C}(\boldsymbol{\theta}')) \nabla \rho(\boldsymbol{\theta}') \mathrm{d}\boldsymbol{\theta}'$$

$$= \mathbb{E}_{\boldsymbol{\theta}' \sim \rho} (\mathbf{A}(\boldsymbol{\theta}') + \mathbf{C}(\boldsymbol{\theta}')) \nabla \log \pi(\boldsymbol{\theta}') k(\cdot, \boldsymbol{\theta}') + \underbrace{\int \nabla \cdot \left( (\mathbf{A}(\boldsymbol{\theta}') + \mathbf{C}(\boldsymbol{\theta}')) k(\cdot, \boldsymbol{\theta}') \right) \mathrm{d}\rho}_{\text{integration by parts}}$$

$$= \mathbb{E}_{\boldsymbol{\theta}' \sim \rho} \Big[ \underbrace{\mathbf{f}(\boldsymbol{\theta}') k(\cdot, \boldsymbol{\theta}')}_{\text{weighted drift coefficient (9)}} + \underbrace{(\mathbf{A} + \mathbf{C})(\boldsymbol{\theta}') \nabla_2 k(\cdot, \boldsymbol{\theta}')}_{\text{repulsive force}} \Big]. \tag{16}$$

$$= \mathbb{E}_{\boldsymbol{\theta}' \sim \rho} \underbrace{\frac{1}{\pi} \nabla \cdot \left( \pi (\mathbf{A} + \mathbf{C}) k(\cdot, \boldsymbol{\theta}') \right)}_{\text{diffusion Stein operator } \mathcal{T}_\pi^{\mathbf{A},\mathbf{C}} k(\cdot, \boldsymbol{\theta}')}, \tag{17}$$

where $\nabla_2$ refers to the gradient w.r.t. the second argument of the kernel function. Interestingly, the kernelized flow $\mathbf{v}_{\mathcal{H}}^{\mathbf{A},\mathbf{C}}$ coincides with calculating the expectation of the diffusion Stein operator $\mathcal{T}_{\pi}^{\mathbf{A},\mathbf{C}}$ (Gorham et al., 2019) applied to the kernel function $k(\boldsymbol{\theta}, \cdot)$, a family of operators mapping functions to zero-expectation functions under an (un-normalized) distribution $\pi$. When $\rho_t$ is approximated by the empirical measure of its particles $\hat{\rho}_t = 1/N \sum_{i=1}^{N} \delta_{\boldsymbol{\theta}_t^{(i)}}$, we can derive the particle update of

$$\dot{\boldsymbol{\theta}}_t = \mathbf{v}_{\mathcal{H}}^{\mathbf{A},\mathbf{C}}(\hat{\rho}_t) = \mathbb{E}_{\boldsymbol{\theta}' \sim \hat{\rho}_t} \mathcal{T}_{\pi}^{\mathbf{A},\mathbf{C}} k(\boldsymbol{\theta}_t, \boldsymbol{\theta}') = \frac{1}{N} \sum_{i=1}^{N} \mathcal{T}_{\pi}^{\mathbf{A},\mathbf{C}} k(\boldsymbol{\theta}_t, \boldsymbol{\theta}_t^{(i)}), \boldsymbol{\theta}_0^{(i)} \sim \rho_0. \quad (18)$$

Similar to SVGD, the particle update of GSVGD leverages between a weighted average of the drift vector and a repulsive force, which drives particles towards the target distribution while preventing a mode collapse. The fiber-gradient Hamiltonian flow on $(\mathcal{P}(\Omega), W_{\mathcal{H}})$ informs a duality between diffusion Stein operator and MCMC dynamics, extending the use of Stein operators beyond a generalization of kernel Stein discrepancy (Gorham et al., 2019).

### 3.3 How does GSVGD approximate MCMC dynamics?

While the derivation of GSVGD originates from the flow interpretation on $W_{\mathcal{H}}$, the analogy between SVGD and GSVGD expresses in equivalent alternative forms discussed by previous literature, further shedding light on the interpretation of GSVGD.

Originally, the SVGD update is expressed in the form of the functional gradient (Liu and Wang, 2016). For MCMC dynamics, we similarly derive the functional gradient with respect to the push-forward measure $\rho_\epsilon = \left( \mathrm{id} + (\mathbf{A} + \mathbf{C})^\top \mathbf{v} \right)_{\#} \rho, \mathbf{v} \in \mathcal{H}^D$,

$$\nabla_{\mathbf{v}} \, \mathrm{KL} \left[ \rho_\epsilon \, \| \, \pi \right] \big|_{\mathbf{v}=\mathbf{0}} = -\mathbb{E}_{\boldsymbol{\theta}' \sim \rho} \mathcal{T}_{\pi}^{\mathbf{A},\mathbf{C}} k(\cdot, \boldsymbol{\theta}') = -\mathbf{v}_{\mathcal{H}}^{\mathbf{A},\mathbf{C}}(\rho). \quad (19)$$

This presents a generalization of the discussion in Liu and Wang (2016) that GSVGD takes incremental transformation of particles in the direction that minimizes KL-divergence, with such transformation warped by $(\mathbf{A}, \mathbf{C})$ and constrained by RKHS.

Alternatively, Liu et al. (2019a) explains SVGD as the projection of the original gradient flow direction $\mathbf{v}(\rho_t) = \nabla \log \pi/\rho_t$ onto an RKHS. We similarly generalize this explanation for SVGD, as it projects $\mathbf{v}^{\mathbf{A},\mathbf{C}}(\rho_t) = (\mathbf{A} + \mathbf{C})\nabla \log \pi/\rho_t \in \mathcal{L}_\rho^2$ onto $\mathcal{H}^D$, as

$$\mathbf{v}_{\mathcal{H}}^{\mathbf{A},\mathbf{C}}(\rho) = \max \cdot \underset{\mathbf{v} \in \mathcal{H}^D, \|\mathbf{v}\|_{\mathcal{H}^D}=1}{\arg\max} \langle \mathbf{v}^{\mathbf{A},\mathbf{C}}(\rho), \mathbf{v} \rangle_{\mathcal{L}_\rho^2}. \quad (20)$$

The analogy of GSVGD goes further when we consider the transportation of $N$ particles as inferring a joint distribution $\pi^{\otimes N}$, which will be covered in the supplements.

### 3.4 GSVGD in practice

We can apply GSVGD updates (16) to de-randomize myriad MCMC dynamics (e.g., Table 1). Notably, GSVGD establishes the previously unexplored duality between Riemannian Langevin diffusion (Girolami and Calderhead, 2011) and Riemannian SVGD (Liu and Zhu, 2017).

To fully harness the capacity of our framework, we can introduce auxiliary (momentum) variables $\mathbf{r}$ to help with the exploration of probability space, namely to augment the target distribution as $\pi(\boldsymbol{\theta}, \mathbf{r}) = \pi(\boldsymbol{\theta})\mathcal{N}(\mathbf{r}|\mathbf{0}, \boldsymbol{\Sigma})$, therefore achieving de-randomized PARVI variant of SGHMC (Chen et al., 2014). Further leveraging a positive definite $\mathbf{G}(\boldsymbol{\theta})$ to efficiently explore the target distribution yields a PARVI variant of SGRHMC (Ma et al., 2015).

When GSVGD is used in conjunction with auxiliary momentum variables, we can employ symmetric splitting for leapfrog-like steps for GSVGDs de-randomizing with momentum,

$$\mathbf{r}_{k+1/2}^{(i)} = \mathbf{r}_k^{(i)} + \frac{\epsilon}{2} \mathbf{v}_{\mathcal{H}}^{\mathbf{A},\mathbf{C}} \left( \mathbf{r}_k^{(i)} \middle| \hat{\rho}(\boldsymbol{\theta}_k, \mathbf{r}_k) \right), \quad (21)$$

$$\boldsymbol{\theta}_{k+1}^{(i)} = \boldsymbol{\theta}_k^{(i)} + \epsilon \mathbf{v}_{\mathcal{H}}^{\mathbf{A},\mathbf{C}} \left( \boldsymbol{\theta}_k^{(i)} \middle| \hat{\rho}(\boldsymbol{\theta}_k, \mathbf{r}_{k+1/2}) \right), \quad (22)$$

$$\mathbf{r}_{k+1}^{(i)} = \mathbf{r}_{k+1/2}^{(i)} + \frac{\epsilon}{2} \mathbf{v}_{\mathcal{H}}^{\mathbf{A},\mathbf{C}} \left( \mathbf{r}_{k+1/2}^{(i)} \middle| \hat{\rho}(\boldsymbol{\theta}_{k+1}, \mathbf{r}_{k+1/2}) \right). \quad (23)$$

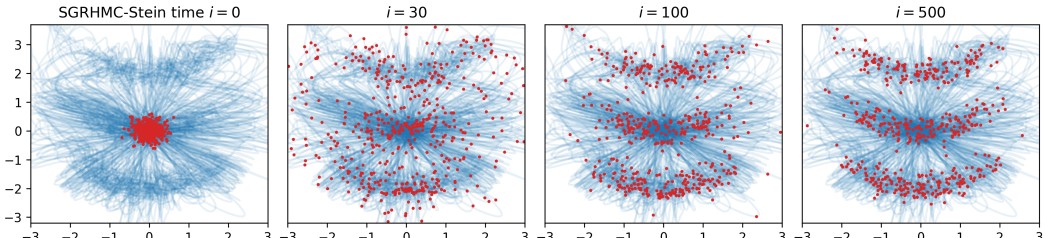

Figure 4: Evolution of RHMC dynamics (blue lines) over four highlighted time increments $k$ (red points) on tri-Gaussian density. The RHMC explores the space by an aggressive outward expansion followed by convergence to the tri-Gaussian density $\pi$.

# 4 Experiments

## 4.1 Stein PARVI on toy data

To demonstrate the efficacy of our methods, we use various PARVI to infer a 3-component mixture of crescent-shaped target measure inspired by Ma et al. (2015). The parametrizations of Riemannian SVGD and SGRHMC-Stein coincide with those in Ma et al. (2015). While all methods explore the nearest mode well (See figure 2), only the SGRHMC-Stein manage to effectively use the Riemannian formulation of $(\mathbf{A}, \mathbf{C})$ to explore the 2 other modes (see figure 3), showcasing the gain in efficiency with application-specific dynamics.

## 4.2 GSVGD on Bayesian neural networks

We apply GSVGD of advanced MCMC dynamics to the inference of Bayesian neural networks, taking a simple structure of one hidden layer, and an output with Gaussian likelihood. We opt for the fully Bayesian specification of BNN, where the precision parameter of the weight prior and the Gaussian likelihood follows Gamma$(1, 0.1)$. The GSVGD variant used in the experiments include SGHMC (Chen et al., 2014) and SGNHT (Ding et al., 2014), with target measure $\pi(\boldsymbol{\theta}, \mathbf{r}, \boldsymbol{\xi}) = \pi(\boldsymbol{\theta})\mathcal{N}(\mathbf{r}|\mathbf{0}, \sigma^2\mathbf{I})\mathcal{N}(\boldsymbol{\xi}|A\mathbf{1}, \mu^{-1}\mathbf{I})$, and $\mathbf{A} = \begin{pmatrix} \mathbf{0} & \mathbf{0} & \mathbf{0} \\ \mathbf{0} & A\mathbf{I} & \mathbf{0} \\ \mathbf{0} & \mathbf{0} & \mathbf{0} \end{pmatrix}, \mathbf{C} = \begin{pmatrix} \mathbf{0} & -\mathbf{I} & \mathbf{0} \\ \mathbf{I} & \mathbf{0} & (\mu\sigma^2)^{-1}\mathrm{diag}(\mathbf{r}) \\ \mathbf{0} & -(\mu\sigma^2)^{-1}\mathrm{diag}(\mathbf{r}) & \mathbf{0} \end{pmatrix}$. As a MCMC form suitable when used in conjunction with stochastic gradients, SGNHT takes additional temperature parameter $\boldsymbol{\xi}$. The log-likelihood results in Table 2 demonstrates that the de-randomized MCMC dynamics achieve superior predictive performance than their LD counterparts, largely thanks to the exploration of probability space from the momentum variables (See figure 5), with a notable gain from symmetric splitting. SGNHT-Stein methods show robustness to hyperparameter selection and stochastic gradient noise.

| Method | boston | concrete | energy | kin8nm | power | yacht | year | protein |
|---|---|---|---|---|---|---|---|---|
| LD | -2.52 (0.15) | -3.16 (0.04) | -2.36 (0.06) | 0.10 (0.03) | -2.85 (0.03) | -1.60 (0.08) | -3.69 (0.01) | -3.05 (0.01) |
| SVGD | -2.60 (0.46) | -3.11 (0.10) | -1.95 (0.07) | 1.03 (0.25) | -2.82 (0.03) | -1.66 (0.17) | -3.61 (0.00) | -2.91 (0.01) |
| Blob | -2.52 (0.32) | -3.19 (0.07) | -1.67 (0.03) | 0.95 (0.03) | -2.81 (0.04) | -1.34 (0.07) | -3.60 (0.00) | -2.93 (0.01) |
| SGHMC-Blob | **-2.42 (0.12)** | **-2.95 (0.01)** | -1.36 (0.26) | 1.23 (0.02) | -2.77 (0.04) | **-0.73 (0.46)** | -3.60 (0.02) | -3.73 (0.16) |
| SGNHT | -2.59 (0.13) | -3.41 (0.05) | -2.44 (0.04) | 0.75 (0.02) | -2.86 (0.02) | -2.64 (0.04) | -3.69 (0.00) | -3.01 (0.01) |
| DE | -2.68 (0.63) | -2.96 (0.15) | **-0.45 (0.14)** | 1.16 (0.02) | -2.80 (0.03) | -1.04 (0.40) | -3.68 (0.00) | -3.00 (0.01) |
| SGHMC-Stein | -2.81 (0.71) | -3.04 (0.25) | -1.40 (0.82) | **1.25 (0.02)** | **-2.76 (0.04)** | -0.86 (0.33) | **-3.59 (0.00)** | -2.90 (0.01) |
| SGNHT-Stein | -2.49 (0.30) | -2.97 (0.11) | **-0.44 (0.10)** | 1.24 (0.03) | -2.78 (0.04) | -0.85 (0.21) | **-3.59 (0.00)** | **-2.85 (0.01)** |

Table 2: The test log-likelihood (higher is better) results for selected datasets in UCI repository. The results are reported in mean (standard deviation) form averaged over 20 runs in the first 6 columns and 6 runs in the last 2, with the best performing model marked in boldface.

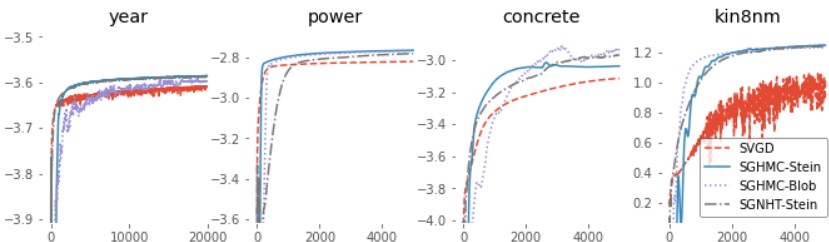

Figure 5: The trace plots of test log-likelihoods of BNN experiments: the x-axis determines the iterations. The figures show the effect of the momentum variables in SGHMC-Stein and SGNHT-Stein in exploring the posterior space, and quickly converging to better optima. The jaggedness of SVGD showcases that while SVGD empirically works with stochastic gradient, it is less robust with stochastic gradient noise.

## 5    Discussion

In this section, we discuss open questions pertinent to the "optimality" of the diffusion Stein operator as a PARVI update, and a direction for future work regarding the fiber-gradient Hamiltonian flow of the chi-squared divergence, inspired by Chewi et al. (2020).

### 5.1    Alternatives to the diffusion Stein operator

The vector field $\mathbf{v}^{\mathbf{A},\mathbf{C}}(\rho) = (\mathbf{A} + \mathbf{C}) \nabla \log \frac{\pi}{\rho}$ is not the unique vector field that induces the time evolution of MCMC dynamics: indeed, two vector fields $\mathbf{v}_1, \mathbf{v}_2$ induce the same curve in $\mathcal{P}(\Omega)$ when they differ by a divergence-free field: $\nabla \cdot (\rho(\mathbf{v}_1(\rho) - \mathbf{v}_2(\rho))) = 0$. Notably, Liu et al. (2019b) propose an alternative vector field $\tilde{\mathbf{v}}^{\mathbf{A},\mathbf{C}}(\rho) = \mathbf{v}^{\mathbf{A},\mathbf{C}}(\rho) - \mathbf{C}\nabla \log \rho + \nabla \cdot \mathbf{C}$ [1]. A projection onto RKHS yields a Stein PARVI with update

$$\dot{\boldsymbol{\theta}}_t = \mathbb{E}_{\rho_t} \left[ \frac{1}{\pi} \nabla \cdot [\pi (\mathbf{A} + \mathbf{C})] k(\boldsymbol{\theta}_t, \cdot) + \mathbf{A}\nabla_2 k(\boldsymbol{\theta}_t, \cdot) \right], \tag{24}$$

consistent with the standard formulation infinitesimal generator. However, the expectation in (24) notably does not converge to zero when $\rho_t = \pi$ (Gorham et al., 2019), suggesting that the particles will keep rotating along the trajectories of a divergence-free vector field, as opposed to stopping, in the equilibrium state. Interested readers can find experiments with respect to this alternative form in the supplementary materials.

### 5.2    Generalizing LAWGD with the diffusion Stein operator

The application of the diffusion Stein operator generalizes Laplacian Adjusted Wasserstein Gradient Descent (LAWGD) (Chewi et al., 2020), which views SVGD as a kernelized Wasserstein gradient flow driven by the chi-squared divergence. Given the first variation of the chi-squared divergence $\frac{\delta \chi^2 [\rho \,\|\, \pi]}{\delta \rho} = \frac{2\rho}{\pi}$, we can verify that the vector field $\mathbf{v}_{\mathcal{H}}^{\mathbf{A},\mathbf{C}} = \mathcal{K}_\rho \left[ (\mathbf{A} + \mathbf{C}) \nabla \log \frac{\pi}{\rho} \right]$ is rewritten as

$$2\mathbf{v}_{\mathcal{H}}^{\mathbf{A},\mathbf{C}} = \mathcal{K}_\pi \left[ (\mathbf{A} + \mathbf{C}) \nabla \frac{2\pi}{\rho} \right] = \mathcal{K}_\pi \left[ (\mathbf{A} + \mathbf{C}) \nabla \frac{\delta \chi^2 [\rho_t \,\|\, \pi]}{\delta \rho_t} \right]. \tag{25}$$

The chi-squared divergence perspectives enriches GSVGD as a kernelized fiber-gradient Hamiltonian flow minimizing the chi-squared divergence. Replacing the vector field $\mathcal{K}_\pi \left[ (\mathbf{A} + \mathbf{C}) \nabla \frac{\mathrm{d}\rho}{\mathrm{d}\pi} \right]$ with $\mathbf{u} = \nabla \cdot \left( \mathcal{K}_\pi \left[ (\mathbf{A} + \mathbf{C}) \frac{\mathrm{d}\rho}{\mathrm{d}\pi} \right] \right)$, we get the dissipation formula for the KL-divergence

$$\frac{\mathrm{dKL}[\rho_t \,\|\, \pi]}{\mathrm{d}t} = -\mathbb{E}_\pi \left\langle \nabla \frac{\rho}{\pi}, \nabla \cdot \left( \mathcal{K}_\pi \left[ (\mathbf{A} + \mathbf{C}) \frac{\rho}{\pi} \right] \right) \right\rangle = -\mathbb{E}_\pi \left[ \frac{\rho}{\pi} \mathcal{A}_\pi^{\mathbf{A},\mathbf{C}} \mathcal{K}_\pi \frac{\rho}{\pi} \right], \tag{26}$$

---

[1]In fact, a divergence-free vector field $\mathbf{C}_0 \nabla \log \rho - \nabla \cdot \mathbf{C}_0$ can be constructed out of an arbitrary skew-symmetric matrix-valued function $\mathbf{C}_0$

where $\mathcal{A}_\pi^{\mathbf{A},\mathbf{C}}$ represents the infinitesimal generator that induces the diffusion Stein operator: $\mathcal{A}_\pi^{\mathbf{A},\mathbf{C}} g = \frac{1}{2\pi}\nabla \cdot (\pi(\mathbf{A} + \mathbf{C})\nabla g)$. Following the construction of LAWGD, we choose the kernel so that $\mathcal{K}_\pi = \left(\mathcal{A}_\pi^{\mathbf{A},\mathbf{C}}\right)^{-1}$. While it remains to be seen what kernel induces this equality, the diffusion Stein operator yields a more flexible formulation, while retaining the superior convergence of LAWGD.

## 6 Conclusion

We further the application of the Stein operator and its generalizations by proposing GSVGD, an interacting particle system transporting a particle set towards a target distribution $\pi$ in an emulation of the corresponding MCMC dynamics. We showcase theoretically and empirically that GSVGD helps augment the well-researched field of efficient MCMC dynamics with deterministic interacting particle systems with high-quality samples.

## Acknowledgments and Disclosure of Funding

We wish to acknowledge the computational resources provided by Aalto Science-IT Project from Computer Science IT and CSC–IT Center for Science, Finland. This work has been supported by the Academy of Finland (Flagship programme: Finnish Center for Artificial Intelligence FCAI and grants no. 292334, 294238, 319264, 334600) and UKRI Turing AI World-Leading Researcher Fellowship, EP/W002973/1.

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
