# Supplementary material to De-randomizing MCMC dynamics with the generalized Stein operator

**Zheyang Shen**[1]    **Markus Heinonen**[1]    **Samuel Kaski**[1,2]
[1]Helsinki Institute for Information Technology, HIIT,
Department of Computer Science, Aalto University, Finland
[2]Department of Computer Science, University of Manchester
{zheyang.shen, markus.o.heinonen, samuel.kaski}@aalto.fi

In the supplementary material, we expand on the background details and derivations mentioned in the paper.

## 1 Additional derivations for the Wasserstein gradient flow

### 1.1 Stochastic differential equations and its Fokker-Planck equation

The general SDE

$$\mathrm{d}\boldsymbol{\theta}_t = \underbrace{\mathbf{f}(\boldsymbol{\theta}_t)}_{\text{drift}}\mathrm{d}t + \underbrace{\sqrt{2\boldsymbol{\Sigma}(\boldsymbol{\theta}_t)}}_{\text{diffusion}}\underbrace{\mathrm{d}\mathbf{W}_t}_{\text{Wiener process}} \tag{1}$$

with an initial distribution $\boldsymbol{\theta}_0 \sim \rho_0$ defines the evolution of a random variable $\boldsymbol{\theta}_t \in \mathbb{R}^D$ over time $t \in \mathbb{R}_+$. The evolution of the marginal distribution $\rho_t$ is given by the Fokker-Planck equation

$$\dot{\rho}_t(\boldsymbol{\theta}) = -\sum_{i=1}^{D}\frac{\partial}{\partial\theta_i}\rho_t(\boldsymbol{\theta})f_i(\boldsymbol{\theta}) + \sum_{i,j=1}^{D}\frac{\partial^2}{\partial\theta_i\partial\theta_j}\rho_t(\boldsymbol{\theta})\Sigma_{ij}(\boldsymbol{\theta}) \tag{2}$$

$$= -\nabla\cdot(\rho_t\mathbf{f}) + (\nabla\nabla):(\rho_t\boldsymbol{\Sigma}), \tag{3}$$

where we use the dyadic vector notation

$$\mathbf{A}:\mathbf{B} = \mathrm{tr}\{\mathbf{A}^\top\mathbf{B}\}. \tag{4}$$

The double-dot notation indicates a sum over all element-wise products, or the sum of all second order partial derivatives. We can verify this with

$$(\nabla\nabla):(\rho_t\boldsymbol{\Sigma}) = \mathrm{tr}\{(\nabla\nabla)^\top(\rho_t\boldsymbol{\Sigma})\} \tag{5}$$

$$= \langle\nabla\nabla, \rho_t\boldsymbol{\Sigma}\rangle_F \tag{6}$$

$$= \sum_{i,j}\frac{\partial^2}{\partial\theta_i\partial\theta_j}\rho_t(\boldsymbol{\theta}_t)\boldsymbol{\Sigma}(\boldsymbol{\theta}_t). \tag{7}$$

### 1.2 FPE of Langevin dynamics

The Langevin dynamics has drift $\mathbf{f}(\boldsymbol{\theta}_t) = \nabla\log\pi(\boldsymbol{\theta}_t)$ and diffusion $\boldsymbol{\Sigma}(\boldsymbol{\theta}_t) = \mathbf{I}$. The Fokker-Planck equation for Langevin dynamics then simplifies into

$$\dot{\rho}_t = -\nabla\cdot(\rho_t\mathbf{f}) + (\nabla\nabla^\top):(\rho_t\mathbf{I}) \tag{8}$$

$$= -\nabla\cdot(\rho_t\nabla\log\pi) + \Delta\rho_t, \tag{9}$$

Preprint. Under review.

where the Laplacian is defined as

$$\Delta \rho_t = \nabla^2 \rho_t = \sum_{i=1}^{D} \frac{\partial^2 \rho_t(\boldsymbol{\theta})}{\partial \theta_i^2}. \tag{10}$$

To derive the FPE we begin by using the Laplacian identity $\Delta \rho_t = \nabla \cdot (\nabla \rho_t)$, resulting in

$$\dot{\rho}_t = \Delta \rho_t - \nabla \cdot \left( \rho_t \nabla \log \pi \right) \tag{11}$$

$$= \nabla \cdot \left( \nabla \rho_t \right) - \nabla \cdot \left( p \nabla \log \pi \right) \tag{12}$$

$$= \nabla \cdot \left( \nabla \rho_t - \rho_t \nabla \log \pi \right) \tag{13}$$

$$= \nabla \cdot \left( \rho_t \nabla \log \rho_t - \rho_t \nabla \log \pi \right) \tag{14}$$

$$= \nabla \cdot \left( \rho_t \nabla \log \frac{\rho_t}{\pi} \right), \tag{15}$$

where we used the identity $\nabla \log \rho = \frac{\nabla \rho}{\rho}$ to expand $\nabla \rho = \rho \nabla \log \rho$. Notice that we did not need to use the common identity $\nabla \cdot (\rho \mathbf{f}) = \rho \nabla \cdot \mathbf{f} + (\nabla \rho) \cdot \mathbf{f}$, nor expand with $\nabla \log \pi = -\nabla H$.

We also notice that the first variation of the Kullback-Leibler divergence has the form $\frac{\delta \mathrm{KL}[\rho_t \parallel \pi]}{\delta \rho_t} = \log \rho_t / \pi + 1$, so that $\nabla \frac{\delta \mathrm{KL}[\rho_t \parallel \pi]}{\delta \rho_t} = \nabla \log \rho_t - \nabla \log \pi$, yielding the final result

$$\dot{\rho}_t = \nabla \cdot \left( \rho_t \nabla \frac{\delta \mathrm{KL}\left[\rho_t \parallel \pi\right]}{\delta \rho_t} \right). \tag{16}$$

In the original variational formulation for the Wasserstein gradient flow (Jordan et al., 1998), the authors prove the weak convergence of a discrete gradient flow taking discrete steps at interval $h$, such that each $\rho_{kh}, k = 1, 2, \ldots$, follows the minimization

$$\rho_{kh} = \underset{\rho \in \mathcal{P}(\Omega)}{\arg \min} \quad \mathrm{KL}\left[\rho \parallel \pi\right] + \frac{1}{2h} W_2^2(\rho, \rho_{(k-1)h}). \tag{17}$$

$(\rho_{kh})_{k=1}^{\infty}$ weakly converges to the Fokker-Planck equation as $h \downarrow 0$.

## 1.3 The Stein-Wasserstein metric

With the Onsager operator defined as $G(\rho)^{-1} : \phi \mapsto -\nabla \cdot (\rho \mathcal{K}_\rho \nabla \phi)$, the Stein-Wasserstein metric between $\rho_0$ and $\rho_1$ is defined using the geometric action function

$$W_{\mathcal{H}}^2(\rho_0, \rho_1) = \inf_{\phi, \rho_t} \left\{ \int_0^1 \int \|\mathcal{K}_{\rho_t} \nabla \phi_t\|_{\mathcal{H}}^2 \, \mathrm{d}t \; : \; \dot{\rho}_t + \nabla \cdot (\rho_t \mathcal{K}_{\rho_t} \nabla \phi_t) = 0 \right\}, \tag{18}$$

where $\mathcal{K}_\rho$ is the integral operator

$$\mathcal{K}_\rho \mathbf{f}(\boldsymbol{\theta}) = \int k(\boldsymbol{\theta}', \boldsymbol{\theta}) \mathbf{f}(\boldsymbol{\theta}') \mathrm{d}\rho(\boldsymbol{\theta}'), \tag{19}$$

which smoothens the function $\mathbf{f}$ over a similar parameters $\boldsymbol{\theta}'$ from the density $\rho$ according to the kernel $k(\boldsymbol{\theta}', \boldsymbol{\theta})$. The Stein-Wasserstein metric follows the definition that the distance between two points in $\mathcal{P}(\Omega)$ consists of the length the shortest arc connecting the two points, parametrized by $\phi_t$.

## 2 Additional derivations for the analysis of MCMC dynamics

To recap, MCMC dynamics consist of positive semi-definite matrix-valued function $\mathbf{A}$ and skew-symmetric matrix-valued function $\mathbf{C}$ with

$$\mathbf{f}(\boldsymbol{\theta}) = \frac{1}{\pi(\boldsymbol{\theta})} \nabla \cdot (\pi(\boldsymbol{\theta})(\mathbf{A}(\boldsymbol{\theta}) + \mathbf{C}(\boldsymbol{\theta}))), \tag{20}$$

$$\boldsymbol{\Sigma}(\boldsymbol{\theta}) = \mathbf{A}(\boldsymbol{\theta}) \tag{21}$$

with matrix properties

$$\mathbf{x}^\top \mathbf{A} \mathbf{x} \geq 0, \qquad \forall \mathbf{x} \in \mathbb{R}^D \tag{22}$$

$$\mathbf{C}^\top = -\mathbf{C}, \quad C_{ij} = -C_{ji}, \quad \mathrm{diag}\,\mathbf{C} = \mathbf{0}. \tag{23}$$

## 2.1 Fokker-Planck equation and its equivalent forms

We derive the Fokker-Planck equation of SDEs with the above MCMC dynamics form as

$$\dot{\rho}_t(\boldsymbol{\theta}) = -\sum_{i=1}^{D} \frac{\partial}{\partial \theta_i} \rho_t(\boldsymbol{\theta}) f_i(\boldsymbol{\theta}) + \sum_{i,j=1}^{D} \frac{\partial^2}{\partial \theta_i \partial \theta_j} \rho_t(\boldsymbol{\theta}) A_{ij}(\boldsymbol{\theta}) \tag{24}$$

$$= -\sum_{i=1}^{D} \frac{\partial}{\partial \theta_i} \rho_t(\boldsymbol{\theta}) f_i(\boldsymbol{\theta}) + \sum_{i,j=1}^{D} \frac{\partial^2}{\partial \theta_i \partial \theta_j} \rho_t(\boldsymbol{\theta}) A_{ij}(\boldsymbol{\theta}) + \underbrace{\sum_{i,j=1}^{D} \frac{\partial^2}{\partial \theta_i \partial \theta_j} \rho_t(\boldsymbol{\theta}) C_{ij}(\boldsymbol{\theta})}_{=0} \tag{25}$$

$$= -\nabla \cdot \left( \frac{\rho_t}{\pi} \nabla \cdot (\pi(\mathbf{A} + \mathbf{C})) \right) + (\nabla\nabla) : (\rho_t(\mathbf{A} + \mathbf{C})) \tag{26}$$

$$= -\nabla \cdot (\rho_t \left[ (\mathbf{A} + \mathbf{C}) \nabla \log \pi(\boldsymbol{\theta}) + \nabla \cdot (\mathbf{A} + \mathbf{C}) \right]) + (\nabla\nabla) : (\rho_t(\mathbf{A} + \mathbf{C})) \tag{27}$$

$$= -\nabla \cdot (\rho_t \left[ (\mathbf{A} + \mathbf{C}) \nabla \log \pi(\boldsymbol{\theta}) + \nabla \cdot (\mathbf{A} + \mathbf{C}) \right]) + \nabla \cdot (\rho_t \left[ (\mathbf{A} + \mathbf{C}) \nabla \log \rho_t + \nabla \cdot (\mathbf{A} + \mathbf{C}) \right]) \tag{28}$$

$$= \nabla \cdot \left( \rho_t(\mathbf{A} + \mathbf{C}) \nabla \log \frac{\rho_t}{\pi} \right), \tag{29}$$

where the skew-symmetric addition is justified via

$$\sum_{i,j=1}^{D} \frac{\partial^2}{\partial \theta_i \partial \theta_j} \rho_t(\boldsymbol{\theta}) C_{ij}(\boldsymbol{\theta}) = \underbrace{\sum_{i,j=1}^{D} \frac{\partial^2 \rho_t(\boldsymbol{\theta})}{\partial \theta_i \partial \theta_j} C_{ij}(\boldsymbol{\theta})}_{=0} + \underbrace{\sum_{i,j=1}^{D} \rho_t(\boldsymbol{\theta}) \frac{\partial^2 C_{ij}(\boldsymbol{\theta})}{\partial \theta_i \partial \theta_j}}_{=0} \tag{30}$$

$$+ \sum_{i,j=1}^{D} \left[ \frac{\partial \rho_t(\boldsymbol{\theta})}{\partial \theta_i} \frac{\partial C_{ij}(\boldsymbol{\theta})}{\partial \theta_j} + \frac{\partial \rho_t(\boldsymbol{\theta})}{\partial \theta_j} \frac{\partial C_{ij}(\boldsymbol{\theta})}{\partial \theta_i} \right] \tag{31}$$

$$= \sum_{i,j=1}^{D} \frac{\partial \rho_t(\boldsymbol{\theta})}{\partial \theta_i} \frac{\partial C_{ij}(\boldsymbol{\theta})}{\partial \theta_j} + \sum_{j,i=1}^{D} \frac{\partial \rho_t(\boldsymbol{\theta})}{\partial \theta_j} \frac{\partial C_{ij}(\boldsymbol{\theta})}{\partial \theta_i} \tag{32}$$

$$= \sum_{i,j=1}^{D} \frac{\partial \rho_t(\boldsymbol{\theta})}{\partial \theta_i} \frac{\partial C_{ij}(\boldsymbol{\theta})}{\partial \theta_j} + \sum_{j,i=1}^{D} \frac{\partial \rho_t(\boldsymbol{\theta})}{\partial \theta_j} \frac{\partial - C_{ji}(\boldsymbol{\theta})}{\partial \theta_i} = 0. \tag{33}$$

The final form (29) corresponds to a generalization of the continuity equation $\dot{\rho}_t + \nabla \cdot (\rho_t(\mathbf{A} + \mathbf{C}) \nabla \phi_t) = 0$.

## 2.2 The diffusion Stein operator

Originally, the Stein's identity (Stein, 1972) maps sufficiently regular functions $\boldsymbol{\Phi} : \mathbb{R}^D \mapsto \mathbb{R}^D$ to $\mathcal{T}\boldsymbol{\Phi}(\boldsymbol{\theta}) = \nabla \log \pi(\boldsymbol{\theta}) \cdot \boldsymbol{\Phi}(\boldsymbol{\theta}) + \nabla \cdot \boldsymbol{\Phi}(\boldsymbol{\theta})$. The function $\mathcal{T}\boldsymbol{\Phi}$ has expectation zero under the target measure $\pi$: $\mathbb{E}_\pi \mathcal{T}\boldsymbol{\Phi} = 0$, yielding the original Stein's identity. As a generalization, Gorham et al. (2019) discuss the application of infinitesimal generator of MCMC dynamics as a means to discover operators sharing the same property. Infinitesimal generators of Feller processes describes the perturbation of functions:

$$\mathcal{A}u(\boldsymbol{\theta}) = \lim_{t \to 0} \frac{\mathbb{E}\left[u(\boldsymbol{\theta}_t)|\boldsymbol{\theta}_0 = \boldsymbol{\theta}\right] - \mathbf{u}(\boldsymbol{\theta})}{t}. \tag{34}$$

For MCMC dynamics with parametrization $(\mathbf{A}, \mathbf{C})$, the infinitesimal generator is calculated as

$$\left( \mathcal{A}_\pi^{\mathbf{A}, \mathbf{C}} u \right)(\boldsymbol{\theta}) = \frac{1}{2\pi(\boldsymbol{\theta})} \nabla \cdot (\pi(\boldsymbol{\theta})(\mathbf{A}(\boldsymbol{\theta}) + \mathbf{C}(\boldsymbol{\theta})) \nabla u(\boldsymbol{\theta})). \tag{35}$$

And the *diffusion Stein operator*, denoted as $\mathcal{T}_\pi^{\mathbf{A}, \mathbf{C}}$ in this work, is defined by substituting $\frac{\nabla u}{2}$ with a vector-valued function $\mathbf{f}$. GSVGD is defined as $\mathbb{E}_\rho \mathcal{T}_\pi^{\mathbf{A}, \mathbf{C}} k(\cdot, \boldsymbol{\theta})$.

## 2.3 GSVGD as functional gradient in RKHS for incremental transformations

Functional gradient for a functional $F[\cdot]$ is defined as such $\nabla_{\mathbf{f}} F[\mathbf{f}]$ that satisfies $F[\mathbf{f} + \epsilon \mathbf{g}] = F[\mathbf{f}] + \langle \nabla_{\mathbf{f}} F[\mathbf{f}], \mathbf{g} \rangle_{\mathcal{H}^D} + O(\epsilon^2)$. The particle update of GSVGD can be seen as the functional gradient with respect to the push-forward measure $\rho_\epsilon = \left( \mathrm{id} + (\mathbf{A} + \mathbf{C})^\top \mathbf{f} \right)_\# \rho = (\mathrm{id} + (\mathbf{A} - \mathbf{C})\mathbf{f})_\# \rho$. Following the proof of Theorem 3.3 in Liu and Wang (2016), we define $F[\mathbf{f}] = \mathrm{KL} [\rho_\epsilon \| \pi] = \mathrm{KL} \left[ (\mathrm{id} + (\mathbf{A} - \mathbf{C})\mathbf{f})_\# \rho \| \pi \right] = \mathrm{KL} \left[ \rho \| (\mathrm{id} + (\mathbf{A} - \mathbf{C})\mathbf{f})_\#^{-1} \pi \right]$

$$F[\mathbf{f} + \epsilon \mathbf{g}] = \mathbb{E}_\rho \left[ \log \rho(\boldsymbol{\theta}) - \log \pi(\boldsymbol{\theta} + (\mathbf{A} - \mathbf{C})(\mathbf{f} + \epsilon \mathbf{g})) - \log \det \left( \mathbf{I} + \underbrace{\nabla((\mathbf{A} - \mathbf{C})(\mathbf{f} + \epsilon \mathbf{g}))}_{\text{Jacobian matrix}} \right) \right]. \tag{36}$$

We then have

$$F[\mathbf{f} + \epsilon \mathbf{g}] - F[\mathbf{f}] = -\underbrace{\mathbb{E}_\rho \left[ \log \frac{\pi(\boldsymbol{\theta} + (\mathbf{A} - \mathbf{C})(\mathbf{f} + \epsilon \mathbf{g}))}{\pi(\boldsymbol{\theta} + (\mathbf{A} - \mathbf{C})\mathbf{f})} \right]}_{\Delta_1} - \underbrace{\mathbb{E}_\rho \left[ \log \frac{\det (\mathbf{I} + \nabla((\mathbf{A} - \mathbf{C})(\mathbf{f} + \epsilon \mathbf{g})))}{\det (\mathbf{I} + \nabla((\mathbf{A} - \mathbf{C})\mathbf{f}))} \right]}_{\Delta_2}. \tag{37}$$

$$\Delta_1 = \mathbb{E}_\rho \left[ \log \pi(\boldsymbol{\theta} + (\mathbf{A} - \mathbf{C})(\mathbf{f} + \epsilon \mathbf{g})) - \log \pi(\boldsymbol{\theta} + (\mathbf{A} - \mathbf{C})\mathbf{f}) \right] \tag{38}$$

$$= \epsilon \mathbb{E}_\rho \left[ \nabla \log \pi(\boldsymbol{\theta} + (\mathbf{A} - \mathbf{C})\mathbf{f}) \cdot (\mathbf{A} - \mathbf{C})\mathbf{g} \right] + O(\epsilon^2) \tag{39}$$

$$= \mathbb{E}_\rho \left[ (\mathbf{A} + \mathbf{C})\nabla \log \pi(\boldsymbol{\theta} + (\mathbf{A}(\boldsymbol{\theta}) - \mathbf{C}(\boldsymbol{\theta}))\mathbf{f}(\boldsymbol{\theta})) \right] \cdot \mathbf{g} + O(\epsilon^2) \tag{40}$$

$$= \mathbb{E}_{\boldsymbol{\theta} \sim \rho} \left[ (\mathbf{A}(\boldsymbol{\theta}) + \mathbf{C}(\boldsymbol{\theta}))\nabla \log \pi(\boldsymbol{\theta} + (\mathbf{A}(\boldsymbol{\theta}) - \mathbf{C}(\boldsymbol{\theta}))\mathbf{f}(\boldsymbol{\theta})) \right] \cdot \langle k(\boldsymbol{\theta}, \cdot), \mathbf{g}(\cdot) \rangle_{\mathcal{H}^D} + O(\epsilon^2) \tag{41}$$

$$= \langle \mathbb{E}_{\boldsymbol{\theta} \sim \rho} \left[ (\mathbf{A}(\boldsymbol{\theta}) + \mathbf{C}(\boldsymbol{\theta}))\nabla \log \pi(\boldsymbol{\theta} + (\mathbf{A}(\boldsymbol{\theta}) - \mathbf{C}(\boldsymbol{\theta}))\mathbf{f}(\boldsymbol{\theta})) \right] k(\boldsymbol{\theta}, \cdot), \mathbf{g}(\cdot) \rangle_{\mathcal{H}^D} + O(\epsilon^2), \tag{42}$$

$$\Delta_2 = \mathbb{E}_\rho \left[ \log \det (\mathbf{I} + \nabla((\mathbf{A} - \mathbf{C})(\mathbf{f} + \epsilon \mathbf{g}))) - \log \det (\mathbf{I} + \nabla((\mathbf{A} - \mathbf{C})\mathbf{f})) \right] \tag{43}$$

$$= \epsilon \mathbb{E}_\rho \left[ (\mathbf{I} + \nabla((\mathbf{A} - \mathbf{C})\mathbf{f}))^{-1} : \nabla ((\mathbf{A} - \mathbf{C})\mathbf{g}) \right] + O(\epsilon^2) \tag{44}$$

$$= \epsilon \mathbb{E}_\rho \left[ (\mathbf{I} + \nabla((\mathbf{A} - \mathbf{C})\mathbf{f}))^{-1} : \left\{ \underbrace{(\mathbf{A} - \mathbf{C}) \nabla \mathbf{g}}_{\Delta_3} + \underbrace{\mathbf{M}}_{\Delta_4} \right\} \right] + O(\epsilon^2), \tag{45}$$

where $\mathbf{M}_{ij} = \sum_{\ell=1}^D \frac{\partial (\mathbf{A} - \mathbf{C})_{i\ell}}{\partial \boldsymbol{\theta}_j} g_\ell$. We have

$$\mathbf{M}_{ij}(\boldsymbol{\theta}) = \sum_{\ell=1}^D \frac{\partial (\mathbf{A}(\boldsymbol{\theta}) - \mathbf{C}(\boldsymbol{\theta}))_{i\ell}}{\partial \boldsymbol{\theta}_j} g_\ell(\boldsymbol{\theta}) \tag{46}$$

$$= \sum_{\ell=1}^D \frac{\partial (\mathbf{A}(\boldsymbol{\theta}) - \mathbf{C}(\boldsymbol{\theta}))_{i\ell}}{\partial \boldsymbol{\theta}_j} \langle k(\boldsymbol{\theta}, \cdot), g_\ell(\cdot) \rangle_{\mathcal{H}} \tag{47}$$

$$= \langle \mathbf{h}^{(ij)} k(\boldsymbol{\theta}, \cdot), \mathbf{g}(\cdot) \rangle_{\mathcal{H}^D}, \tag{48}$$

where $\mathbf{h}_\ell^{(ij)} = \frac{\partial(\mathbf{A}(\boldsymbol{\theta}) - \mathbf{C}(\boldsymbol{\theta}))_{i\ell}}{\partial\boldsymbol{\theta}_j}$.

$$\Delta_3 = \epsilon\mathbb{E}_\rho\left[(\mathbf{I} + \nabla((\mathbf{A} - \mathbf{C})\mathbf{f}))^{-1} : \{(\mathbf{A} - \mathbf{C})\,\nabla\mathbf{g}\}\right] \tag{49}$$

$$= \epsilon\mathbb{E}_\rho\left[(\mathbf{I} + \nabla((\mathbf{A} - \mathbf{C})\mathbf{f}))^{-1}(\mathbf{A} + \mathbf{C}) : \nabla\mathbf{g}\right] \tag{50}$$

$$= \epsilon\mathbb{E}_{\boldsymbol{\theta}\sim\rho}\left[(\mathbf{I} + \nabla((\mathbf{A}(\boldsymbol{\theta}) - \mathbf{C}(\boldsymbol{\theta}))\mathbf{f}(\boldsymbol{\theta})))^{-1}(\mathbf{A}(\boldsymbol{\theta}) + \mathbf{C}(\boldsymbol{\theta})) : \langle\nabla_1 k(\boldsymbol{\theta}, \cdot), \mathbf{g}(\cdot)\rangle_{\mathcal{H}^D}\right] \tag{51}$$

$$= \epsilon\left\langle\mathbb{E}_{\boldsymbol{\theta}\sim\rho}\left[(\mathbf{I} + \nabla((\mathbf{A}(\boldsymbol{\theta}) - \mathbf{C}(\boldsymbol{\theta}))\mathbf{f}(\boldsymbol{\theta})))^{-1}(\mathbf{A}(\boldsymbol{\theta}) + \mathbf{C}(\boldsymbol{\theta}))\nabla_1 k(\boldsymbol{\theta}, \cdot), \mathbf{g}(\cdot)\right]\right\rangle_{\mathcal{H}^D}, \tag{52}$$

$$\Delta_4 = \epsilon\mathbb{E}_{\boldsymbol{\theta}\sim\rho}\left[\underbrace{(\mathbf{I} + \nabla((\mathbf{A}(\boldsymbol{\theta}) - \mathbf{C}(\boldsymbol{\theta}))\mathbf{f}(\boldsymbol{\theta})))^{-1}}_{\mathbf{Q}(\boldsymbol{\theta})} : \mathbf{M}\right] \tag{53}$$

$$= \epsilon\mathbb{E}_{\boldsymbol{\theta}\sim\rho}\sum_{i,j=1}^D \mathbf{Q}_{ij}\mathbf{M}_{ij} = \epsilon\left\langle\mathbb{E}_{\boldsymbol{\theta}\sim\rho}\sum_{i,j=1}^D \mathbf{Q}_{ij}\mathbf{h}^{(ij)}k(\boldsymbol{\theta}, \cdot), \mathbf{g}(\cdot)\right\rangle_{\mathcal{H}^D} \tag{54}$$

It is straightforward that when $\mathbf{f} = \mathbf{0}$,

$$\Delta_4 = \epsilon\mathbb{E}_{\boldsymbol{\theta}\sim\rho}\left\langle\sum_{i=1}^D \mathbf{h}^{(ii)}k(\boldsymbol{\theta}, \cdot), \mathbf{g}\right\rangle_{\mathcal{H}^D}, \tag{55}$$

$$\sum_{i=1}^D \mathbf{h}_\ell^{(ii)} = \sum_{i=1}^D \frac{\partial(\mathbf{A} - \mathbf{C})_{i\ell}}{\partial\boldsymbol{\theta}_i} \tag{56}$$

$$= \sum_{i=1}^D \frac{\partial(\mathbf{A} + \mathbf{C})_{\ell i}}{\partial\boldsymbol{\theta}_i} = \nabla\cdot(\mathbf{A} + \mathbf{C}). \tag{57}$$

Using the definition of $\Delta_i, i \in [4]$, we can derive the GSVGD particle update as $\mathbf{f} = \mathbf{0}$, confirming the functional derivative coinciding with the GSVGD particle update.

## 2.4 Projection onto RKHS

To prove $\mathbf{v}_\mathcal{H}^{\mathbf{A},\mathbf{C}}$ is the projection of $\mathbf{v}^{\mathbf{A},\mathbf{C}}$ onto $\mathcal{H}^D$, we start from the inner product on $\mathcal{L}_\rho^2$, such that $\forall\mathbf{v} \in \mathcal{H}^D$

$$\langle\mathbf{v}^{\mathbf{A},\mathbf{C}}(\cdot|\rho), \mathbf{v}\rangle_{\mathcal{L}_\rho^2} = \mathbb{E}_\rho\left[(\mathbf{A} + \mathbf{C})\nabla\log\pi/\rho\cdot\mathbf{v}\right] \tag{58}$$

$$= \mathbb{E}_\rho\left[(\mathbf{A} + \mathbf{C})\nabla\log\pi\cdot\mathbf{v}\right] - \mathbb{E}_\rho\left[(\mathbf{A} + \mathbf{C})\nabla\log\rho\cdot\mathbf{v}\right] \tag{59}$$

$$= \mathbb{E}_\rho\left[(\mathbf{A} + \mathbf{C})\nabla\log\pi\cdot\mathbf{v}\right] - \underbrace{\int\left[(\mathbf{A} + \mathbf{C})\nabla\rho\cdot\mathbf{v}\right]\mathrm{d}\boldsymbol{\theta}}_{\text{weak derivative of measures}} \tag{60}$$

$$= \mathbb{E}_\rho\left[(\mathbf{A} + \mathbf{C})\nabla\log\pi\cdot\mathbf{v}\right] + \mathbb{E}_\rho\left[\sum_{i,j}\frac{\partial((\mathbf{A}_{ij} + \mathbf{C}_{ij})v_i)}{\partial\boldsymbol{\theta}_j}\right] \tag{61}$$

$$= \mathbb{E}_\rho\left[\{(\mathbf{A} + \mathbf{C})\nabla\log\pi + \nabla\cdot(\mathbf{A} + \mathbf{C})\}\cdot\mathbf{v}\right] + \mathbb{E}_\rho\left[\sum_{i,j}\frac{(\mathbf{A}_{ij} + \mathbf{C}_{ij})\partial v_i}{\partial\boldsymbol{\theta}_j}\right] \tag{62}$$

$$= \left\langle\mathbb{E}_{\boldsymbol{\theta}\sim\rho}\{(\mathbf{A}(\boldsymbol{\theta}) + \mathbf{C}(\boldsymbol{\theta}))\nabla\log\pi(\boldsymbol{\theta}) + \nabla\cdot(\mathbf{A}(\boldsymbol{\theta}) + \mathbf{C}(\boldsymbol{\theta}))\}k(\boldsymbol{\theta}, \cdot), \mathbf{v}(\cdot)\right\rangle_{\mathcal{H}^D} \tag{63}$$

$$+ \mathbb{E}_\rho\left[\sum_{i,j}\frac{(\mathbf{A}_{ij} + \mathbf{C}_{ij})\partial v_i}{\partial\boldsymbol{\theta}_j}\right], \tag{64}$$

$$\mathbb{E}_\rho \left[ \sum_{i,j} \frac{(\mathbf{A}_{ij} + \mathbf{C}_{ij})\partial v_i}{\partial \boldsymbol{\theta}_j} \right] = \mathbb{E}_{\boldsymbol{\theta} \sim \rho} \left[ \sum_{i,j} (\mathbf{A}_{ij}(\boldsymbol{\theta}) + \mathbf{C}_{ij}(\boldsymbol{\theta})) \frac{\partial v_i(\boldsymbol{\theta})}{\partial \boldsymbol{\theta}_j} \right] \tag{65}$$

$$= \mathbb{E}_{\boldsymbol{\theta} \sim \rho} \left[ \sum_{i,j} (\mathbf{A}_{ij}(\boldsymbol{\theta}) + \mathbf{C}_{ij}(\boldsymbol{\theta})) \langle \frac{\partial}{\partial \boldsymbol{\theta}_j} k(\boldsymbol{\theta}, \cdot), \mathbf{v}_i(\cdot) \rangle_{\mathcal{H}} \right] \tag{66}$$

$$= \left\langle \mathbb{E}_{\boldsymbol{\theta} \sim \rho} \left[ (\mathbf{A} + \mathbf{C}) \nabla k(\boldsymbol{\theta}, \cdot) \right], \mathbf{v}(\cdot) \right\rangle_{\mathcal{H}^D}. \tag{67}$$

Therefore, confirming that $\forall \mathbf{v} \in \mathcal{H}^D$, $\langle \mathbf{v}^{\mathbf{A},\mathbf{C}}, \mathbf{v} \rangle_{\mathcal{L}_\rho^2} = \langle \mathbf{v}_{\mathcal{H}}^{\mathbf{A},\mathbf{C}}, \mathbf{v} \rangle_{\mathcal{H}^D}$.

## 2.5 Interpreting GSVGD as MCMC dynamics of $\pi^{\otimes N}$

Additionally, we can view GSVGD with with constant $\mathbf{C}$ matrices and $N$ particles as the mean-field limit of a MCMC dynamics inferring the product target measure $\underbrace{\pi \times \ldots \times \pi}_{N} = \pi^{\otimes N}$, a rather trivial extension of the discussion in (Gallego and Insua, 2018). The $\mathbf{A}(\boldsymbol{\theta}^{\otimes N}), \mathbf{C}(\boldsymbol{\theta}^{\otimes N}) \in \mathbb{R}^{ND \times ND}$ is defined as

$$\tilde{\mathbf{A}}_{i,j} = \frac{1}{N} k(\boldsymbol{\theta}_i, \boldsymbol{\theta}_j) \frac{\mathbf{A}(\boldsymbol{\theta}_i) + \mathbf{A}(\boldsymbol{\theta}_j)}{2}, \tag{68}$$

$$\tilde{\mathbf{C}}_{i,j} = \frac{1}{N} k(\boldsymbol{\theta}_i, \boldsymbol{\theta}_j) \left[ \mathbf{C} + \frac{\mathbf{A}(\boldsymbol{\theta}_j) - \mathbf{A}(\boldsymbol{\theta}_i)}{2} \right]. \tag{69}$$

We can verify that the MCMC dynamics

$$\dot{\boldsymbol{\theta}}_t^{\otimes N} = \frac{1}{\pi(\boldsymbol{\theta}^{\otimes N})} \nabla \cdot \left( \pi(\boldsymbol{\theta}_t^{\otimes N}) \left( \tilde{\mathbf{A}}(\boldsymbol{\theta}_t^{\otimes N}) + \tilde{\mathbf{C}}(\boldsymbol{\theta}_t^{\otimes N}) \right) \right) + \sqrt{2\tilde{\mathbf{A}}(\boldsymbol{\theta}_t^{\otimes N})} d\mathbf{W}_{ND}, \tag{70}$$

takes the invariant measure $\pi^{\otimes N}$. And that the drift coefficient corresponds to the GSVGD particle update. Furthermore, this framework accepts non-constant $\mathbf{C}$ matrices when $\tilde{\mathbf{A}}_{i,j} = \frac{1}{N} k(\boldsymbol{\theta}_i, \boldsymbol{\theta}_j) \left[ \frac{\mathbf{A}(\boldsymbol{\theta}_i) + \mathbf{A}(\boldsymbol{\theta}_j)}{2} + \frac{\mathbf{C}(\boldsymbol{\theta}_j) - \mathbf{C}(\boldsymbol{\theta}_i)}{2} \right]$ remains positive semidefinite.

It is worth noting that as $N \to \infty$, the drift coefficient goes to $\mathbf{0}$, making GSVGD the mean-field limit of such dynamics.

## 2.6 Stochastic particle optimization sampling (SPOS) as MCMC dynamics

Viewing GSVGD as the mean-field limit of MCMC dynamics yields additional insights. For example, Zhang et al. (2020) propose stochastic particle optimization sampling in the form of

$$\dot{\boldsymbol{\theta}}_t^{\otimes N} = \frac{1}{\pi(\boldsymbol{\theta}_t^{\otimes N})} \nabla \cdot \left( \pi(\boldsymbol{\theta}_t^{\otimes N}) \left( \overline{\mathbf{K}} \otimes \mathbf{I} + \sigma^2 \mathbf{I} \right) \right) + \sqrt{2\sigma^2 \mathbf{I}} \mathbf{W}_{ND}, \tag{71}$$

where $\overline{\mathbf{K}}_{ij} = k(\boldsymbol{\theta}_i, \boldsymbol{\theta}_j)$. Formally, SPOS combines the SVGD particle update with a step of (multi-chained) Langevin diffusion. The SPOS particle update does not conform to the standard formulation of MCMC dynamics, yielding a biased sampling algorithm. However, such bias can be fixed by changing the diffusion coefficient into $\sqrt{2 \left( \overline{\mathbf{K}} \otimes \mathbf{I} + \sigma^2 \mathbf{I} \right)}$. Translating into discretized dynamics, SPOS generates samples from the target measure when the injected noise is correlated across particles. However, the correlated injected noise has variance approaching zero as $N \to \infty$.

## 2.7 Recovering SVGD with GFSF

While SVGD takes the form of gradient flow on $(\mathcal{P}(\Omega), W_{\mathcal{H}})$, we can connect SVGD with other form of smoothing discussed in Liu et al. (2019a), noted as gradient flow with smoothed test functions (GFSF). GSVGD taking $\mathbf{A} = \overline{\mathbf{K}} \otimes \mathbf{I}, \mathbf{C} = \mathbf{0}$ gives

$$\mathbf{v}^{\mathbf{A},\mathbf{C}} = \overline{\mathbf{K}} \otimes \mathbf{I} \nabla \log \pi^{\otimes N} / \rho^{\otimes N} = \overline{\mathbf{K}} \otimes \mathbf{I} \nabla \log \pi^{\otimes N} - \overline{\mathbf{K}} \otimes \mathbf{I} \nabla \log \rho^{\otimes N}. \tag{72}$$

Using the GFSF estimation of $\nabla \log \rho$ [1], the term $\overline{\mathbf{K}} \otimes \mathbf{I} \nabla \log \rho^{\otimes N}$ yields $\nabla \cdot \overline{\mathbf{K}}$, recovering the SVGD particle update. Viewing PARVI in the product space yields additional insights of connecting different smoothing methods. Analogously, we can use the generalized Stein's identity (Gorham et al., 2019) to arrive at a GFSF smoothing of GSVGD.

## 3 Additional discussion

### 3.1 Formulating Riemannian Langevin diffusion as gradient flow on $(\mathcal{P}(\Omega), W_{2,\mathbf{A}})$

With $\mathbf{C} = \mathbf{0}$ and positive definite $\mathbf{A}$, we can generalize the 2-Wasserstein metric in the Benamou-Brenier form (Benamou and Brenier, 2000)

$$W_{2,\mathbf{A}}^2(\rho_0, \rho_1) = \inf_{\phi, \rho_t} \left\{ \int_0^1 \int \langle \nabla \phi_t, \mathbf{A} \nabla \phi_t \rangle \mathrm{d}t : \dot{\rho}_t + \nabla \cdot (\rho_t \mathbf{A} \nabla \phi_t) = 0 \right\}. \tag{73}$$

The Onsager operator of $W_{2,\mathbf{A}}$ takes the form $G(\rho)^{-1} : \phi \mapsto -\nabla \cdot (\rho \mathbf{A} \nabla \phi)$. MCMC dynamics such as Riemannian Langevin diffusion (Girolami and Calderhead, 2011) can be interpreted as a gradient flow of KL $[\rho \,\|\, \pi]$ on $W_{2,\mathbf{A}}$: $\dot{\rho}_t = \nabla \cdot \left( \rho_t \mathbf{A} \nabla \frac{\delta \mathrm{KL}[\rho_t \,\|\, \pi]}{\delta \rho_t} \right) = -G(\rho_t)^{-1} \frac{\delta \mathrm{KL}[\rho_t \,\|\, \pi]}{\delta \rho_t}$, circumventing the necessity of defining a projection $\mathfrak{p}_\rho$ onto the tangent space of $(\mathcal{P}(\Omega), W_2)$.

### 3.2 How to accelerate PARVI for underdamped Langevin diffusion?

Ma et al. (2019) argue that the underdamped Langevin diffusion with $\mathbf{A} = \begin{pmatrix} \mathbf{0} & \mathbf{0} \\ \mathbf{0} & A\mathbf{I} \end{pmatrix}, \mathbf{C} = \begin{pmatrix} \mathbf{0} & -\mathbf{I} \\ \mathbf{I} & \mathbf{0} \end{pmatrix}$ is an analog of Nesterov's acceleration of the overdamped Langevin diffusion $\mathbf{A} = \mathbf{I}, \mathbf{C} = \mathbf{0}$, and such analog still stands for their PARVI variants. Similar to results presented in MCMC research (Mou et al., 2021), we can construct PARVI with third-order Langevin diffusion $\mathbf{A} = \begin{pmatrix} \mathbf{0} & \mathbf{0} & \mathbf{0} \\ \mathbf{0} & \mathbf{0} & \mathbf{0} \\ \mathbf{0} & \mathbf{0} & A\mathbf{I} \end{pmatrix}, \mathbf{C} = \begin{pmatrix} \mathbf{0} & -\mathbf{I} & \mathbf{0} \\ \mathbf{I} & \mathbf{0} & -\gamma\mathbf{I} \\ \mathbf{0} & \gamma\mathbf{I} & \mathbf{0} \end{pmatrix}$, equivalent to applying a higher-order momentum method in gradient descent.

### 3.3 Momentum resampling

One unexplored aspect of particle variational inference with momentum variable is the possibility of turning PARVI into a proper sampling algorithm, one feat unattainable by de-randomization of LD, as deterministic optimization of $N$ particle can only produce $N$ samples. With the introduction of momentum variables, it is possible to periodically resample the momentum variable to obtain more samples from the target distribution – in practice, it involves combining the deterministic particle updates with a jump process that routinely samples from the marginal distribution of momentum.

Aside from possibly obtaining more samples, resampling during the optimization can also speed up convergence to the target distribution. As we know the marginal distribution with respect to $\mathbf{r}$, resampling of momentum variables reduces the KL-divergence between $\rho_t$ and $\pi$, as KL $[\rho_t(\boldsymbol{\theta})\pi(\mathbf{r}) \,\|\, \pi(\boldsymbol{\theta})\pi(\mathbf{r})] \leq$ KL $[\rho_t(\theta, \mathbf{r}) \,\|\, \pi(\boldsymbol{\theta})\pi(\mathbf{r})]$. It remains a theoretical and empirical open question whether resampling momentum can speed up convergence.

## 4 Experiment details

### 4.1 Toy experiments

The 2-dimensional likelihood of the toy experiments used in this paper takes the form of $\pi \propto \frac{1}{3} \sum_{i=1}^3 \exp\left( -\frac{x^4}{10} + \frac{(z_i y - x^2)^2}{2} \right)$, $z_i = \{-2, 0, 2\}$, and the original particle loca-

---

[1] formally, GFSF is equivalent of applying the Stein gradient estimator (Li and Turner, 2017) without regularization.

tions are initialized as $\begin{pmatrix} x \\ y \end{pmatrix} \sim \mathcal{N}(0, 0.01\mathbf{I})$. We can use the energy function $U(\boldsymbol{\theta}) = \log\left(\frac{1}{3}\sum_{i=1}^{3}\exp\left(-\frac{x^4}{10} + \frac{(z_i y - x^2)^2}{2}\right)\right)$ to construct Riemannian samplers in the experimental setting consistent with Ma et al. (2015), where Fisher information metric matrix is defined as $\mathbf{G}^{-1}(\boldsymbol{\theta}) = D\sqrt{|U(\boldsymbol{\theta}) + C|}, U = 1.5, C = 0.5$. In Riemannian Stein variational gradient descent (SVGD) (Liu and Zhu, 2017), we follow the practice of Riemannian LD (Girolami and Calderhead, 2011) and parametrize $\mathbf{A}(\boldsymbol{\theta}) = \mathbf{G}^{-1}(\boldsymbol{\theta}), \mathbf{C}(\boldsymbol{\theta}) = \mathbf{0}$; In Riemannian SGHMC, we follow (Ma et al., 2015) parametrize $\mathbf{A}(\boldsymbol{\theta}) = \begin{pmatrix} \mathbf{0} & \mathbf{0} \\ \mathbf{0} & \mathbf{G}^{-1}(\boldsymbol{\theta}) \end{pmatrix}, \mathbf{C}(\boldsymbol{\theta}) = \begin{pmatrix} \mathbf{0} & -\mathbf{G}^{-1/2} \\ \mathbf{G}^{-1/2} & \mathbf{0}(\boldsymbol{\theta}) \end{pmatrix}$.

## 4.2 Bayesian neural network experiments

In Bayesian neural network for regression, we use a standard structure of 1 hidden layers with width 50, along with a conjugate prior on the precision parameter of its weight priors. Specifically, we have

$$y \sim \mathcal{N}\left(\mathbf{W}_2^\top \text{ReLU}(\mathbf{W}_1^\top \mathbf{x} + \mathbf{b}_1) + \mathbf{b}_2, \gamma^{-1}\right), \tag{74}$$

$$\mathbf{W}_1, \mathbf{b}_1, \mathbf{w}_2, \mathbf{b}_2 \sim \mathcal{N}(\mathbf{0}, \lambda^{-1}\mathbf{I}), \mathbf{W}_1 \in \mathbb{R}^{D \times 50}, \mathbf{b}_1 \in \mathbb{R}^{50}, \tag{75}$$

$$\gamma, \lambda \sim \text{Gamma}(1, 0.1). \tag{76}$$

The weights $\mathbf{W}_i$ are initialized with glorot normal distribution and $\mathbf{b}_i$ are intialized with zero.

The hyperparameters for the methods applied in the paper are selected by cross-validation in the following fashion: the learning rate $\eta$ is selected in $\eta \in \{10^{-8}, 10^{-7}, 10^{-6}, 10^{-5}, 10^{-4}, 10^{-3}, 10^{-2}\}$; for methods involving additional momentum variables $\mathbf{r}$, we adopt the momentum interpretation of the underdamped Langevin dynamics (Chen et al., 2014) and select momentum term $\alpha \in \{0.01, 0.1, 0.5\}$ [2]; for thermostat-type samplers, we additionally tune the precision parameter of the temperature variable, with $\mu \in \{0.1, 1.0, 10.0\}$. For methods involving kernel parameters, we parameterize a squared exponential kernel with the median method (Liu and Wang, 2016). We take symmetric splitting for methods involving momentum variables.

For the 6 datasets from UCI repository, we take a $90\%/10\%$ training/test partition of the data; for the 6 medium-sized datasets (except for "year" and "protein"), we take 20 different training / test splits, 6 splits for "protein", and 6 different initializations with the "year" dataset, as the split is fixed. We implemented the models using JAX (Bradbury et al., 2018), and ran the experiments on Nvidia Volta V100 GPU nodes. We present the technical formulation of the methods and its running time in Table 1 and Table 2, respectively.

| Method | A | C | $\pi$ | hyperparameters | Reference |
|---|---|---|---|---|---|
| LD | $\mathbf{I}$ | $\mathbf{0}$ | $\pi(\boldsymbol{\theta})$ | $\eta = \epsilon$ | Welling and Teh (2011) |
| SVGD | $\mathbf{I}$ | $\mathbf{0}$ | $\pi(\boldsymbol{\theta})$ | $\eta = \epsilon$ | Liu and Wang (2016) |
| Blob | $\mathbf{I}$ | $\mathbf{0}$ | $\pi(\boldsymbol{\theta})$ | $\eta = \epsilon$ | Chen et al. (2018) |
| DE | - | - | $\pi(\boldsymbol{\theta})$ | $\eta = \epsilon, \alpha$ | Lakshminarayanan et al. (2017) |
| SGHMC-Blob | $\begin{pmatrix} \mathbf{0} & \mathbf{0} \\ \mathbf{0} & A\mathbf{I} \end{pmatrix}$ | $\begin{pmatrix} \mathbf{0} & -\mathbf{I} \\ \mathbf{I} & \mathbf{0} \end{pmatrix}$ | $\pi(\boldsymbol{\theta})\mathcal{N}(\mathbf{r}|\mathbf{0}, \sigma^2\mathbf{I})$ | $\eta = \epsilon^2\sigma^{-2}, \alpha = \epsilon\sigma^{-2}A$ | Liu et al. (2019b) |
| SGHMC-Stein | $\begin{pmatrix} \mathbf{0} & \mathbf{0} \\ \mathbf{0} & A\mathbf{I} \end{pmatrix}$ | $\begin{pmatrix} \mathbf{0} & -\mathbf{I} \\ \mathbf{I} & \mathbf{0} \end{pmatrix}$ | $\pi(\boldsymbol{\theta})\mathcal{N}(\mathbf{r}|\mathbf{0}, \sigma^2\mathbf{I})$ | $\eta = \epsilon^2\sigma^{-2}, \alpha = \epsilon\sigma^{-2}A$ | this work |
| SGNHT | $\begin{pmatrix} \mathbf{0} & \mathbf{0} & \mathbf{0} \\ \mathbf{0} & A\mathbf{I} & \mathbf{0} \\ \mathbf{0} & \mathbf{0} & \mathbf{0} \end{pmatrix}$ | $\begin{pmatrix} \mathbf{0} & -\mathbf{I} & \mathbf{0} \\ \mathbf{I} & \mathbf{0} & (\mu\sigma^2)^{-1}\text{diag}(\mathbf{r}) \\ \mathbf{0} & -(\mu\sigma^2)^{-1}\text{diag}(\mathbf{r}) & \mathbf{0} \end{pmatrix}$ | $\pi(\boldsymbol{\theta})\mathcal{N}(\mathbf{r}|\mathbf{0}, \sigma^2\mathbf{I})\mathcal{N}(\boldsymbol{\xi}|A\mathbf{1}, \mu^{-1}\mathbf{I})$ | $\eta = \epsilon^2\sigma^{-2}, \alpha = \epsilon\sigma^{-2}A, \mu$ | Ding et al. (2014) |
| SGNHT-Stein | $\begin{pmatrix} \mathbf{0} & \mathbf{0} & \mathbf{0} \\ \mathbf{0} & A\mathbf{I} & \mathbf{0} \\ \mathbf{0} & \mathbf{0} & \mathbf{0} \end{pmatrix}$ | $\begin{pmatrix} \mathbf{0} & -\mathbf{I} & \mathbf{0} \\ \mathbf{I} & \mathbf{0} & (\mu\sigma^2)^{-1}\text{diag}(\mathbf{r}) \\ \mathbf{0} & -(\mu\sigma^2)^{-1}\text{diag}(\mathbf{r}) & \mathbf{0} \end{pmatrix}$ | $\pi(\boldsymbol{\theta})\mathcal{N}(\mathbf{r}|\mathbf{0}, \sigma^2\mathbf{I})\mathcal{N}(\boldsymbol{\xi}|A\mathbf{1}, \mu^{-1}\mathbf{I})$ | $\eta = \epsilon^2\sigma^{-2}, \alpha = \epsilon\sigma^{-2}A, \mu$ | this work |

Table 1: An overview of the parameters in the methods used in the BNN experiments paper, including the (possibly augmented) target distribution, the parameterizations of MCMC dynamics, and the tunable hyperparameters (step size $\eta$, momentum term $\alpha$ and precision term for the temperature variable $\mu$).

## 4.3 Additional experiments

Apart from the standard from of GSVGD and Blob methods in the paper, we experimented Bayesian neural network with particle-based variational inference (PARVI) consistent with the pSGHMC-det

---

[2]It is notable that the step size $\epsilon$ in the discretization of dynamics does not directly correspond to the "effective learning rate" in SGHMC-type samplers.

|  | boston | concrete | energy | kin8nm | power | yacht | year | protein |
|---|---|---|---|---|---|---|---|---|
| LD | 124.88(22.68) | 91.40(45.34) | 98.17(41.45) | 131.89(21.67) | 131.01(14.20) | 86.91(42.83) | 811.08(140.66) | 559.76(9.84) |
| SVGD | 75.76(10.04) | 75.31(13.82) | 67.70(19.69) | 87.03(3.68) | 81.98(10.59) | 71.11(10.14) | 704.39(30.75) | 348.16(16.67) |
| Blob | 79.69(2.25) | 75.39(14.36) | 66.79(19.86) | 83.55(10.36) | 75.35(18.10) | 66.94(18.52) | 708.06(16.48) | 352.89(11.15) |
| HMC-Blob | 76.76(20.62) | 80.95(19.51) | 83.31(16.53) | 93.61(13.63) | 87.99(16.13) | 73.33(22.05) | 874.68(165.39) | 409.75(13.78) |
| SGNHT | 231.86(55.61) | 245.63(25.54) | 252.53(3.79) | 256.84(2.74) | 232.18(60.56) | 222.90(59.43) | 1326.01(197.56) | 1016.57(79.80) |
| DE | 76.76(10.52) | 75.53(14.57) | 75.30(7.04) | 80.76(13.80) | 79.26(15.56) | 70.80(13.89) | 709.06(23.05) | 351.96(13.21) |
| SGHMC-Stein | 89.03(10.60) | 91.82(8.58) | 85.50(13.45) | 95.02(9.94) | 90.87(9.73) | 81.02(16.11) | 761.57(119.60) | 395.49(14.61) |
| SGNHT-Stein | 101.22(3.27) | 94.33(13.16) | 89.52(18.69) | 86.18(19.88) | 96.86(14.21) | 89.98(13.62) | 879.78(14.91) | 429.61(13.22) |

Table 2: Mean (standard deviation) of running times for BNN experiments measured in seconds. All methods are run for 5,000 iterations (first 6 columns, averaged over 20 runs) and 20,000 iterations (last 2 columns, averaged over 6 runs), respectively.

formula in Liu et al. (2019b), and the corresponding Stein version with reproducing kernel Hilbert space (RKHS) projection. The experiment results do not show clear difference from the standard form. From the perspective of Hamiltonian dynamics, we can view underdamped Langevin dynamics (LD) and this variant of PARVI both as Hamiltonian Monte Carlo with a "continuous resampling" of the momentum variable: underdamped LD resamples momentum by running overdamped LD on the momentum variable; its PARVI variant runs SVGD (Stein) or Blob variant as continuous resampling. While the particles do not converge to an equilibrium, the marginal distribution $\rho_t$ remains unchanged.

|  | boston | concrete | energy | kin8nm | power | yacht | year | protein |
|---|---|---|---|---|---|---|---|---|
| SGHMC-Blob* | 11.68(39.93) | 0.21(0.45) | 0.00(0.00) | 0.07(0.00) | 0.05(0.00) | 0.00(0.00) | 0.64(0.00) | 0.49(0.01) |
| SGHMC-Stein* | 0.12(0.07) | 0.09(0.02) | 0.00(0.00) | 0.07(0.00) | 0.05(0.00) | 0.00(0.00) | 0.65(0.00) | 0.48(0.01) |

Table 3: Mean (standard deviation) of mean squared error with PARVI experiment: the experimental setting is the same as the standard BNN experiment.

|  | boston | concrete | energy | kin8nm | power | yacht | year | protein |
|---|---|---|---|---|---|---|---|---|
| SGHMC-Blob* | -2.58(0.12) | -2.93(0.08) | -0.49(0.10) | 1.23(0.02) | -2.77(0.04) | -0.70(0.36) | -3.58(0.00) | -2.87(0.01) |
| SGHMC-Stein* | -2.54(0.28) | -2.98(0.09) | -0.32(0.24) | 1.25(0.02) | -2.78(0.03) | -0.76(0.53) | -3.59(0.00) | -2.87(0.01) |

Table 4: Mean (standard deviation) of text log-likelihood with PARVI experiment: the experimental setting is the same as the standard BNN experiment.