# OpenReview forum: "De-randomizing MCMC dynamics with the diffusion Stein operator"
_NeurIPS.cc/2021/Conference — NeurIPS 2021 Poster_

### Official Review · Reviewer_FarY · 2021-07-01

**Rating:** 5
**Confidence:** 3

**Summary:**

The paper combines the results of two previous works: (Liu, 2017) and (Liu et al., 2019). I'll first describe the contribution of the previous works and then will move to the contribution of the current paper.

(Liu, 2017) provides the theoretical analysis of the Stein Variational Gradient Descent (SVGD). It shows that the evolution of the density under SVGD is described by the Vlasov equation (non-linear analog of the Fokker-Planck equation); moreover, in the space of distributions, SVGD defines the gradient descent minimizing the KL divergence between the current point and the target density (similarly, Langevin Dynamics {LD} defines the gradient descent in the Wasserstein space). Importantly, to derive this result (Liu, 2017) introduces a novel metric in the distribution space (H-Wasserstein distance) based on the Stein operator.

In a similar fashion, (Liu et al., 2019) provides the analysis of different MCMC dynamics (Langevin, HMC, RMHMC, SGHMC) by considering the time evolution of the densities in the space of distributions. They first show how any MCMC dynamics can be reformulated as a completely deterministic procedure by deriving the corresponding vector field in the state space, which induces a vector field in the Wasserstein space. Further, they derive a kind of a Hodge decomposition for the vector field in the Wasserstein space, which provides many insights into existing MCMC dynamics. To describe SGHMC and SG Nose-Hoover thermostat algorithms, the authors introduce the Fiber-Riemannian manifold, which is a Fiber Bundle, but with the Riemannian structure on each fiber. Unlike (Liu, 2017), this paper works in the Wasserstein space (distributions with a finite second momentum equipped with the 2-Wasserstein distance).

The current paper goes the same way as (Liu et al., 2019), but uses the H-Wasserstein distance (introduced in {Liu, 2017}) for the space of distributions. To be more precise, it starts with the same family of MCMC dynamics (introduced in (Ma et al., 2015)) and then uses Theorem 5 from (Liu et al., 2019) changing the projection operator onto the tangent space, which changes because of the different metric. Substituting the different metric into Theorem 5 from (Liu et al., 2019) the authors extend the previous results by additionally describing the SGRHMC algorithm.

- (Liu, 2017) Qiang Liu. Stein variational gradient descent as gradient flow. In Advances in Neural Information Processing Systems, volume 30, pages 3115–3123, 2017.
- (Liu et al., 2019) Chang Liu, Jingwei Zhuo, and Jun Zhu. Understanding MCMC dynamics as flows on the Wasserstein space. In International Conference on Machine Learning, pages 4093–4103, 2019.
- (Ma et al., 2015) Yi-An Ma, Tianqi Chen, and Emily B. Fox. A complete recipe for stochastic gradient MCMC. In NIPS’15 Proceedings of the 28th International Conference on Neural Information Processing Systems - Volume 2, volume 28, pages 2917–2925, 2015.

**Limitations And Societal Impact:**

The limitations of the paper are not addressed properly because of the weak comparison against the previous works.

**Main Review:**

I would describe the theoretical contribution of the paper as incremental. The authors marginally improve the developments from (Liu et al., 2019) by including one more algorithm into the "derandomization framework". Although this inclusion is of course a positive thing, I would classify the paper as practical, targeting the derandomization of the concrete MCMC dynamics (SGRHMC {Stochastic Gradient Riemannian HMC}). Since it is not clear (and not discussed in the paper) which metric provides better convergence, both metrics must be compared empirically (despite that H-Wasserstein includes 2-Wasserstein). I recall that the metrics yield different derandomization schemes of MCMC dynamics.

The empirical study in the paper is unsatisfactory. The main implication of the proposed framework is the derandomization of SGRHMC, however, the empirical results for this algorithm are demonstrated only for a toy example. Since the proposed framework is built upon the results of (Liu et al., 2019) it must be compared against it where it is possible. However, the convergence plots (Figure 5) compare the proposed algorithm only against SVGD. The comparison in Table 2 is hard to understand since the names of some methods are not described in the text. I guess, that SGHMC-Blob corresponds to a method from (Liu et al., 2019) and it performs comparably to the proposed SGHMC-Stein. I would also suggest comparing the proposed framework to both derandomization schemes from (Liu et al., 2019): equations 7 and 11.

I think the clarity of the paper could be significantly improved. From the beginning, the paper tricks the reader into thinking that it develops the whole framework of MCMC derandomization from scratch. To avoid this possible confusion I would suggest the authors explicitly mention that they start their developments from (Liu et al., 2019), which is already describing a large body of MCMC dynamics. In general, the structure of the paper is not easy to follow, additionally hindered by many minor flows. I will mention some appearance flaws, which should be considered as suggestions:
1. Figure 1. "discretization" under the second arrow is not so clear since we can discretize in space, time, particles.
2. Figure under Table 1 is not illustrative, the only thing it says is that LD and HMC are stochastic and the rest are deterministic particles. I think the target audience of the paper knows it.
3. The notation for the matrix scalar product (colon in the text) is not introduced in the main text.
4. Notations E, x, H^D are not introduced in the main text.
5. The derivation of the FP equation in section 3.1 is superfluous for further developments.
6. The formulation of the tangent space (line 165) is important for the result. Therefore, the definition of H^D here would help the reader a lot.

**Time Spent Reviewing:**

15 (I was reading previous papers and I'm not so fluent in differential geometry on probabilistic manifolds)

---

> ### Author Response · Authors · 2021-08-10
> **Reviewer FarY: a discussion on this work's contribution to Stein's method, and an open question about zero-divergence flows**
>
> Thank you for your insightful review, as well as your time and effort, prompting us to take a critical look to better our paper. In our response, we mainly address issues delineated by the review, namely (i) the contribution of this paper when compared to Liu et al. [1] and how [1] is credited in the paper; (ii) the experiments section.
>
> (i) Our paper builds on the premise of [1], but presents a simplified extended framework that harnesses a general class of Stein operators, which compare favorably with Blob method used in [1]. In addition, we present a more complete view on the application of Stein's methods by exploring the interpretation of GSVGD. We think that the new perspectives outlined in the paper warrant a significant contribution.
>
> We present a particle variational inference framework that accepts a broader class of MCMC dynamics by simplifying the Fokker-Planck equation. The first step in developing GSVGD is to add a term of zero value to the Fokker-Planck equation of MCMC dynamics (eq. 12), an algebraic step independent from the fiber-gradient Hamiltonian (fGH) flow viewpoint. Essentially, the ODE $\dot{\theta}_t = (A+C)\nabla\log\pi(\theta_t)/\rho_t(\theta_t)$ always coincides in FP equation with the Ito diffusion parametrized by A and C. This demonstrates that particle variational inference exist beyond assumption 4 (i.e., a restriction on A sharing a block-diagonal structure, and C conforming to the requirements of a Hamiltonian flow) in [1], which SGNHT-type samplers notably violate, as well as samplers that do not conform to A and C matrices sharing block-diagonal factorizing structures [6]. While SGRHMC is used as an example, we propose a framework that accepts all proper MCMC dynamics that converge to the target distribution, which also leads to a revelation of a previously unexplored duality between Riemannian variants of Langevin dynamics and SVGD.
>
> Stein-based particle variational inference is more robust to kernel choices and exhibits more straightforward theoretical properties. Despite being labelled as equivalent options [2], Stein and Blob methods respond differently to kernel parameters, due to the estimation of $\nabla\log\rho_t$. Blob method directly mimics the fGH flow in 2-Wasserstein space with a kernel-based estimate of the gradient of logdensity. However, applying Stein's method does not require such estimates, as it is implicitly handled inside the expectation term (section 2.4 in the supplements) from weak derivative of measures. An integral part of PARVI with Blob method, its particle updates critically depend on reliable gradient estimates, and suboptimal ones lead to inaccurate samples. A clear demonstration of this drawback can be seen from Figure 3 in [2], when poor choice of kernel leads to samples collapsing into modes, while SVGD samples do not. The importance of gradient estimation is amplified by more complex MCMC dynamics, as a one-estimate-fits-all approach is agnostic to MCMC parameters. The dependency of gradient estimation makes it difficult to study theoretical properties of Blob-based PARVI in general, while numerous papers [3-4] study the theory of SVGD, which often extends into GSVGD.
>
> Besides establishing GSVGD from kernelizing the fGH flow, we also thoroughly study the connection to the broad class of Stein's method, which completes the picture of its application. While Gorham et al. [5] established a connection between MCMC dynamics and Stein operators, it remained unclear what the application of the diffusion Stein operator in the context of PARVI entails. We propose two alternative viewpoints that extend existing interpretations of SVGD in 3.3. We believe that such study sheds light on a holistic picture of both PARVI and Stein's method at large.
>
> The reviewer has also noted that the language in the paper "tricks the reader" into overblown claims of contribution. We respectfully disagree with this statement. It is often stated in the text that the formulation of GSVGD "follows" and "adapts" existing framework of [1], and in the passage posing the central research question (L37), we immediately credited [1] for a preliminary solution, while stating in the next paragraph that the fGH flow is adapted for a new metric.
>
> (ii) We appreciate your suggestions of improving the clarity of the experiment section and will work towards a clearer explanation, especially with clear statements of the naming conventions of methods. We will also include the traceplot curves from SGHMC-Blob in figure 5 as a comparison.
>
> We have a few thoughts about eqs. 7 and 11 in [1], which leads to an important open question in PARVI de-randomization of MCMC dynamics. In our experiments, we used SGHMC-Blob to denote eq. 11 in [1] (originally named pSGHMC-fGH), and you have suggested that we also compare to eq. 7 (pSGHMC-det). [1] proposes two alternatives of PARVI because they induce the same Fokker-Planck equation as MCMC dynamics. However, there exist an infinite number of ODEs that fit the description. For example, we could use an arbitrary skew-symmetric matrix-valued function $C_0$, and formulate an ODE sharing the same FP equation: $\dot{\theta}_t = (A+C)\nabla\log\pi(\theta_t)/\rho_t(\theta_t) + C_0\nabla\log\rho_t(\theta_t)+\nabla\cdot C_0$. This is because the added vector field has zero divergence, i.e., $\nabla\cdot(\rho (C_0\nabla\log\rho+\nabla\cdot C_0))=0$. Notably pSGHMC-det differs from pSGHMC-fGH with $C_0=C$. The choice of vector fields is an interesting topic for future study, however, we opted for pSGHMC-fGH for the fact that it induces a zero vector when $\rho=\pi$, conforming to the intuition of particle variational inference as an iterative optimization procedure. We will clarify this motivation in the paper.
>
> However, applying pSGHMC-det has its upsides, as its particle update more closely resembles gradient descent with momentum. We could use the same kernelization trick to obtain a "Stein-like" PARVI (which in fact corresponds to an alternative infinitesimal generator for MCMC dynamics, mentioned by Gorham et al. [5], footnote 2), and we denote the methods as SGHMC-Blob* and SGHMC-Stein*, respectively. The experiment results, which we report the RMSE and test log-likelihood results in the two tables below, demonstrate no clear preference compared to methods used in the paper based on pSGHMC-fGH, showcasing the necessity to study the effect of zero-divergence vector fields in future work.
>
> |             | boston       | concrete   | energy     | kin8nm     | power      | yacht      | year       | protein    |
> |:------------|:-------------|:-----------|:-----------|:-----------|:-----------|:-----------|:-----------|:-----------|
> | SGHMC-Blob*  | 11.68(39.93) | 0.21(0.45) | 0.00(0.00) | 0.07(0.00) | 0.05(0.00) | 0.00(0.00) | 0.64(0.00) | 0.49(0.01) |
> | SGHMC-Stein* | 0.12(0.07)   | 0.09(0.02) | 0.00(0.00) | 0.07(0.00) | 0.05(0.00) | 0.00(0.00) | 0.65(0.00) | 0.48(0.01) |
>
> |             | boston      | concrete    | energy      | kin8nm     | power       | yacht       | year        | protein     |
> |:------------|:------------|:------------|:------------|:-----------|:------------|:------------|:------------|:------------|
> | SGHMC-Blob*  | -2.58(0.12) | -2.93(0.08) | -0.49(0.10) | 1.23(0.02) | -2.77(0.04) | -0.70(0.36) | -3.58(0.00) | -2.87(0.01) |
> | SGHMC-Stein* | -2.54(0.28) | -2.98(0.09) | -0.32(0.24) | 1.25(0.02) | -2.78(0.03) | -0.76(0.53) | -3.59(0.00) | -2.87(0.01) |
>
> We appreciate the suggestions made in the detailed comments, and will briefly address them below, marked consistently with the order:
>
> 1. The vagueness of "discretization" in figure 1: The word is left without description because practical implementations of Wasserstein flow involve multiple types of discretization. We will include explanatory texts in the annotation of the figure.
> 3. We understand the current lack of context in the use of colon, and will revise the text to include a description.
> 4. We will add brief descriptions that note that functional $E$ in the Wasserstein gradient flow is an energy functional minimized through the gradient flow, and detail that the notation $\mathcal{H}^D$ denotes the RKHS of an operator-valued kernel defined given a scalar kernel $k$, specifically $\mathbf{k}(x, x') = k(x, x') \mathbf{I}$.
> 5. Eqs. 11-13 are important derivations to showcase that the fGH flow can be constructed from the FP equation, and we are working on to-the-point explanation for the derivation.
> 6. Besides clarifying the notation of $\mathcal{H}^D$, we will include reasoning in the derivation stating that the developments are derived using the reproducing property of operator-valued RKHS.
>
> [1] Liu, Chang, Jingwei Zhuo, and Jun Zhu. "Understanding MCMC dynamics as flows on the Wasserstein space." International Conference on Machine Learning. PMLR, 2019.
>
> [2] Liu, Chang, et al. "Understanding and accelerating particle-based variational inference." International Conference on Machine Learning. PMLR, 2019.
>
> [3] Duncan, Andrew, Nikolas Nusken, and Lukasz Szpruch. "On the geometry of Stein variational gradient descent." arXiv preprint arXiv:1912.00894 (2019).
>
> [4] Liu, Qiang, and Dilin Wang. “Stein Variational Gradient Descent as Moment Matching.” Advances in Neural Information Processing Systems, vol. 31, 2018, pp. 8854–8863.
>
> [5] Gorham, Jackson, et al. "Measuring sample quality with diffusions." The Annals of Applied Probability 29.5 (2019): 2884-2928.
>
> [6] Futami, Futoshi, et al. “Accelerating the Diffusion-Based Ensemble Sampling by Non-Reversible Dynamics.” ICML 2020: 37th International Conference on Machine Learning, vol. 1, 2020, pp. 3337–3347.

---

> > ### Comment · Reviewer_FarY · 2021-08-21
> > **after response**
> >
> > Thank you for the detailed response! Unfortunately, the response doesn't address my main concerns. To be sure, I've gone through the paper one more time.
> >
> > I think I need to highlight my concerns about the significance of the contribution.
> >
> > (i)
> > - I don't see how this paper simplifies the Fokker-Planck equation, and I don't see this as a contribution since eq. (13) is the same equation as derived in (Ma et al., 2015).
> > - Moreover, the proposed vector field (eq. 14) differs from the vector field from (Liu et al., 2019) only by the definition of the projector operator. That's why I find the main development to be incremental.
> > - I understand that it allows describing all Ito processes (preserving target), as well as it doesn't rely on the estimation of the current density (unlike (Liu et al., 2019)). However, the framework of (Liu et al., 2019) also describes Ito processes deterministically. Also, as mentioned in the paper, the proposed vector field could estimate the current density implicitly by working in RKHS. I agree that the proposed vector field is neater, though.
> > - I understand, that the proposed vector field is connected to SVGD (as described in 3.3), and this connects MCMC and SVGD. However, either do (Liu et al., 2019a from the paper) and (Liu, 2017).
> >
> > Overall, I, think that the authors propose a neat way to write several things, but I still think that the paper oversells its contribution (especially in the abstract).
> >
> > (ii)
> > The empirical study still is of great concern for me. By the way, the results for SGHMC-Blob in the author's response are much better for the "energy" dataset than in the paper. Based, on the provided results I see that the proposed approach performs on par with the approach of (Liu et al., 2019), which should be made explicit in the text. This fact, probably, hints that the change of the projection operator doesn't contribute much to the performance.
> >
> > To conclude I would like to raise my score to be closer to the consensus, but I still think that the paper (in its current form) is below the acceptance threshold. I hope my unconfident opinion wouldn't raise many problems for the authors.

---

### Official Review · Reviewer_Kj2w · 2021-07-10

**Rating:** 7
**Confidence:** 3

**Summary:**

The authors propose a generalization of Stein Variational Gradient
Descent (SVGD) that hopes to produce more reliable surrogate measures
of an invariant measure pi. SVGD can be viewed as a procedure
that is minimizing the KL divergence by taking (functional) gradient
steps in an RKHS; it can similarly be viewed as an approximation to
the gradient flow induced by Langevin dynamics by accommodating H^d
as its function space. This paper studies the generalization of SVGD
that occurs by replacing Langevin dynamics by the dynamics induced
from an Ito diffusion with invariant measure pi. Using Ito diffusions
in lieu of the overdamped Langevin diffusion produces deterministic
analogs of SGRLD, SGHMC and SGRHMC. This also permits the
formulation of GSVGD, a variant of SVGD that incorporates auxiliary
variables in a flavor similar to HMC. The authors demonstrate the
efficacy of this PARVI method on a toy 3-component GMM and on
Bayesian neural nets for a suite of datasets.

**Limitations And Societal Impact:**

No, it might be helpful to draw out how these ideas could have a negative
impact on research, e.g., could these methods lead researchers astray?

**Main Review:**

The main contribution of the paper is the development of GSVGD: a
generalization of SVGD that applies the mechanics of any Ito process
(rather than the overdamped Langevin diffusion). There are some nice
connections to other related concepts, namely a view of this under
a Wasserstein gradient flow. Overall, I thought the paper was a nice
read that laid out some complicated concepts clearly. My biggest
critique of the paper is the limited empirical evidence; it is great
to see the method working on a toy model, but it would be great to
add another experiment demonstrating its efficacy.

SVGD tends to lead to under-dispersed examples in lower dimensions (D < 50)
and I wonder if those same limitations apply here. Given the benefit
of this method is a way to harness both a Riemannian structure and also
momentum variables, I would also like to see a comparison of this method
with SGRHMC. I'm curious how much the determinism here hurts, as MCMC
methods tend to work well in high dimensional inference.

Originality:
This is mostly an extension of previous work, so it not completely novel. But the application of this to other operators is a significant improvement.

Validity:
The claims in the paper look well-founded and correct.

Clarity:
The paper was clearly written and easy to follow. There were a few notational issues; see the detailed comments below.

Significance:
If this yielded a more accurate version of SVGD, that could be a significant improvement for PARVI samplers.

Detailed comments:

Eqn 2: It might be helpful to define the notation (grad^T grad) : (rho_t I)
explicitly somewhere.

L80: "Given the property of differential operators": which property are you
referring to?

Eqn 6: What's v_H
L183: One might consider citing "Minimum Stein Discrepancy Estimators" by
Barp et al. as another use of the diffusion Stein operator for the purposes
of estimation.

L215-L220: At some point, the authors should explain how G(theta) is
defined, as this is used in the Riemannian methods.

Table 2: Are LD and SGNHT the only stochastic methods benchmarked? Why
was SGHMC not experimented with? I'm curious how the deterministic methods
compare to the randomized ones.

**Time Spent Reviewing:**

3

---

> ### Author Response · Authors · 2021-08-10
> **Reviewer Kj2w: on the dispersion of SVGD samples**
>
> Thank you for the positive review, as well as your time and effort.
>
> Thank you for bringing up the interesting point that deterministic dynamics might suffer from under-dispersed samples in high dimensions. While it definitely is a sensible hypothesis and warrants further analysis, we think that specific drawbacks of PARVI algorithms require further theoretical and empirical work, and a well-informed conclusion cannot be drawn at this point beyond anecdotal evidence from toy experiments.
>
> Here are our thoughts on the dispersion of SVGD samples. To summarize the previous work by Liu and Wang [1], SVGD tends to converge to a fixed-point solution with particle updates being zero. Such solutions depend on the choice of kernels. [1] explores this topic in detail, demonstrating that there exist a "Stein matching set", that is, a set of functions exactly estimated by the fixed-point solution. The Stein matching set analysis naturally extends to the fixed points of GSVGDs, as we can replace the Langevin Stein operator in [1] with diffusion Stein operators. Therefore, how the samples disperse depends on the interplay between the kernel and parametrization of the diffusion Stein operator. Following standard practice of previous works, we use RBF kernel where the lengthscale parameter is selected heuristically by the median method. This heuristic is susceptible to initialization, and can over- or under-disperse the samples in its convergence. Kernel selection is PARVI is very much an open question apart from some preliminary findings (for example, the linear kernel guarantees exact estimates of first and second moments of the target distribution with fixed-point SVGD solutions [1]), and there is newly explored room to improve due to the enriched set of Stein operators. We need to leave such study of GSVGD to future work.
>
> We will briefly answer the questions in the detailed comments below, in the same order:
>
> (i) Definition of the $\nabla^\top\nabla$ notation: We agree that this notation causes confusion, and we will revise the text with clear explanation and change to $\nabla\nabla$ (thanks to the suggestion of reviewer ubFs for better consistency of notations), so as to fit into the paradigm of dyadic notations in the text.
>
> (ii) Differential operators: the sentence in L80 indeed requires clarification, which we will include in the revision. Specifically, second-order differential operators can be written in the form of two divergence operators $\nabla\nabla:Z = \nabla\cdot(\nabla\cdot Z)$, which combines the first term of the Fokker-Planck equation with the second term.
>
> (iii) We appreciate the suggestion to include Barp et al., 2019 as a citation, given that the paper discusses the diffusion Stein operator and has referential value.
>
> (iv) The definition of $G(\theta)$: We have conducted the toy example experiment largely using the same setting as Ma et al. [2], and for a target distribution defined as $\pi\propto\exp(-U(\theta))$, we define $G(\theta)^{-1}=D\sqrt{U(\theta)+C}$, with $D=1.5, C=0.5$. We will include this specific detail in the supplementary materials.
>
> (v) Experiments with SGHMC: in the experiment section, we used SGNHT as a substitution of SGHMC given the numerous moving parts in SGHMC implementations. Myriad works have proposed adaptive procedures to determine the preconditioning matrix and the estimate of the stochastic gradient covariance matrix of SGHMC, and such procedures matter greatly to its performance. In contrast, SGNHT eliminates the need for estimation by introducing auxiliary temperature variables, which we believe is a better fit for comparison.
>
> (vi) Negative impact on research: we believe one should scrutinize the pros and cons of GSVGD when applying to specific Bayesian inference tasks, just like we need to be cautious of choosing MCMC samplers. Despite SVGD being advocated as a "general-purpose" tool for inference, its behaviors have not been fully understood, e.g., the kernel choice issue we have addressed in this response.
>
> [1] Liu, Qiang, and Dilin Wang. “Stein Variational Gradient Descent as Moment Matching.” Advances in Neural Information Processing Systems, vol. 31, 2018, pp. 8854–8863.
>
> [2] Ma, Yi-An, et al. “A Complete Recipe for Stochastic Gradient MCMC.” NIPS’15 Proceedings of the 28th International Conference on Neural Information Processing Systems - Volume 2, vol. 28, 2015, pp. 2917–2925.

---

### Official Review · Reviewer_BVWr · 2021-07-13

**Rating:** 7
**Confidence:** 4

**Summary:**

This paper proposes a generalization of the Stein variational gradient descent (SVGD) algorithm via a discretization of the fiber-gradient Hamiltonian flow. The approach that the authors take to extend the framework of particle-based variational inference is similar to the way in which MCMC dynamics are extended through the "complete recipe" framework in Ma et al. (2015). The authors provide background material to establish the connection between MCMC dynamics, the Fokker-Planck equation and gradient flow methods. The GSVGD then follows by applying the fiber-gradient Hamiltonian flow ideas from MCMC dynamics to the Stein-Wasserstein metric, which is then discretized to produce the GSVGD algorithm. The authors test the efficacy of their algorithm on a toy data model and a Bayesian neural network.

**Limitations And Societal Impact:**

Limitations are partially discussed and no societal impact issues are raised, nor are there expected to be any societal impact issues from this work.

**Main Review:**

Originality – This paper offers an interesting extension to existing work within the SVGD literature. The authors clearly highlight how this work builds upon previous papers and how it differs from published works. The approach taken by the authors, i.e. via the same conceptual route as the complete recipe framework for stochastic gradient MCMC, allows the authors to produce a natural extension of existing particle-based variational inference algorithms and establish connections between existing algorithms.
Quality – The paper appears to be technically sound with accurate derivations. The mathematical and experimental claims are well-supported in the paper. However, the simulation study is perhaps a little short compared to other similar papers. In particular, it would have been nice if the authors were able to identify particular machine learning models, aside from toy models, where the GSVGD approach would be expected to perform better than existing algorithms. Altogether, this is a complete piece of work that offers a nice contribution to the existing literature, while also providing interesting avenues for future development.
Clarity – The paper is very well-written, and the structure of the paper is well-organized. The authors provide a thorough and concise review of existing and related techniques which fit into their framework. The paper provides enough detail for a reader to recreate this work and further derivations provided in the Supplementary Material provide the extra steps needed to fill in the gaps in any of the derivations from the main paper.
Significance – The mathematical results provided in the paper are significant and are very likely to be used by other researchers to develop this line of research further. Due to the limited experimental results, it isn’t clear at this stage whether the GSVGD framework will lead to significant empirical improvements in approximate Bayesian inference, however, the results provided so far, and the experience from the MCMC dynamics community, would suggest that this new class of algorithm will be beneficial.


**Time Spent Reviewing:**

3 hours

---

> ### Author Response · Authors · 2021-08-10
> **Reviewer BVWr: on the applicability of GSVGD**
>
> Thank you for your positive review and we appreciate your insights into our paper, as well as your time and effort. In our response, we briefly address the applicability of GSVGD you brought up in the review.
>
> We believe that the full extent of the application of GSVGD remains to be explored. Given the wide application of diffusion-based MCMC dynamics, we could use their deterministic GSVGD variants as a replacement for drawing efficient, high-quality samples from the target distribution. GSVGD leverages the higher sample quality of deterministic SVGD with flexible Riemannian parametrization of MCMC, so that we could apply application-specific deterministic dynamics to emulate the effective exploration of the target distribution, while maintaining high sample qualities.

---

> > ### Comment · Reviewer_BVWr · 2021-09-01
> > **Response**
> >
> > Thank you for responding to my questions.

---

### Official Review · Reviewer_ubFs · 2021-07-18

**Rating:** 7
**Confidence:** 4

**Summary:**

In this paper the authors generalise the Stein Variational Gradient Descent (SVGD) algorithm by considering an alternative flow, essentially equipping the Stein Geometry with a Riemmanian Poisson structure.   This enables a particle evolution scheme which is no longer a gradient flow of the KL divergence in the Stein geometry.

The resulting flow differs from the original SVGD flow through the introduction of an "irreversible term" characterised by a Skew-Symmetric matrix $C$.

Adopting this approach, the authors are able to consider augmented variable strategies to improve the performance of SVGD by promoting superior exploration, using analogues of underdamped Langevin and Nose-Hoover type dynamics.   This is demonstrated on some toy examples and on a Bayesian Neural Network model.


**Ethical Concerns:**

There are no obvious ethical concerns

**Limitations And Societal Impact:**

They have adequately addressed these.

**Main Review:**

Overall, I find this paper to be an interesting contribution which provides a new generalisation of SVGD.  From a theoretical point of view, this is a close analogy to previous works such as:

[Liu, Chang, Jingwei Zhuo, and Jun Zhu. "Understanding mcmc dynamics as flows on the wasserstein space." International Conference on Machine Learning. PMLR, 2019.]
where a similar programme is undertaken for the Wasserstein gradient flows.

Following on that paper as a parallel study, I feel it was important to study the implication of different choices of the matrices A and C.  In particular, we know that for Langevin gradient flows, degenerate choices of $A$ are possible (both zero or just having zero eigenvalues) are possible, and can yield ergodic dynamics in the right setting, e.g. HMC.   It would be have been interest for the authors to study (at least) numerically the influence of the degeneracy of the matrix $A$, as there are no theoretical results to suggest that $\pi$ is the unique target distribution for SVGD (in the mean field limit).

The calculations in the supplementary materials appear to be correct, albeit formal.   In section 2.5 the authors study the mean-field limit of the system of SDEs associated with SVGD, however they provide no justification (formal or otherwise) of the mean field limit.  The authors should provide this, or at least point the readers to a paper where something similar is done for the standard SVGD case.

Minor details:
Throughout:   The authors frequently write the diffusion part of the Fokker-Planck equation as $\nabla^\top \nabla : (...)$.   Wouldn't $\nabla^\top \nabla$ contract the two gradients to give a Laplacian, in which case the tensor contraction : doesn't make sense?   Wouldn't it be $\nabla \nabla : Z = \sum_{i,j} \frac{\partial^2}{dx_i dx_j} Z_{ij}$ ?





**Needs Ethics Review:**

Yes

**Time Spent Reviewing:**

2

---

> ### Author Response · Authors · 2021-08-10
> **Reviewer ubFs response: A discussion on the degeneracy of A matrix and mean-field interpretation**
>
> Thank you for your positive review and we appreciate your insights into our paper, as well as your time and effort. In our response, we address the three issues you brought up, namely: (i) MCMC dynamics with degenerate $A$ matrices; (ii) the mean-field interpretation of GSVGD; and (iii) the inconsistent notation from the Fokker-Planck equation.
>
> (i) Two different types of degeneracy of the $A$ matrix are mentioned in the review, namely (a) $A=0$; and (b) a nonzero $A$ with zero eigenvalues. We believe that we have adequately addressed (b), considering that the examples of MCMC dynamics used in the paper (apart from overdamped Langevin dynamics) have block-diagonal As, and therefore have zero eigenvalues. As for $A=0$, we explain the reasoning of its absence below.
>
> While one can construct MCMC samplers with $A=0$, they are notably out of the scope of "MCMC dynamics" discussed in the paper. We define MCMC dynamics as such differential equations that yield the target distribution as their stationary distributions, and Hamiltonian Monte Carlo does not fit into the description, because the Hamiltonian equation only yields asymptotically exact samples when combined with a jump process (i.e., momentum resampling). Liu et al. [1] included HMC as a member of MCMC dynamics, but noted its inherent incompatibility with particle variational inference: the fiber-gradient Hamiltonian flow with $A=0$ preserves, instead of minimizes KL-divergence. Building on this insight, we conclude that the Hamiltonian differential equation alone (without a jump process) is not sufficient as MCMC dynamics, and does not yield sensible PARVI algorithms.
>
> (ii) About the mean-field interpretation of (G)SVGD: According to our knowledge, this interpretation originates from Gallego and Rios [2]. To summarize, [2] finds a parametrization of MCMC dynamics whose drift term coincides with the update of SVGD. Therefore, SVGD (and by extension GSVGD) can be seen as simulating an MCMC sampling dynamics (in the form of an SDE) with the diffusion term dropped, hence the usage of the term "mean-field". The dropping of the diffusion term is justified as the diffusion coefficient goes to zero as the number of particles goes to infinity. In the non-asymptotic case, this demonstrates that SVGD with finite particles does yield stationary distribution different from $\pi$, confirming what was mentioned in the reviews. However, the stationary distribution of SVGD coincides with the target distribution in calculating expectation of a "Stein matching class" of functions (see e.g., Liu and Wang [3] for discussion). We will include a few additional sentences of explanation in the supplement.
>
> (iii) The notation used in the Fokker-Planck equation has caused confusion, and we agree that $\nabla\nabla:Z$ is consistent with the dyadic convention used in the paper. We will revise in favor of this notation, while explicitly explain its meaning in the text.
>
> [1] Liu, Chang, Jingwei Zhuo, and Jun Zhu. "Understanding MCMC dynamics as flows on the Wasserstein space." International Conference on Machine Learning. PMLR, 2019.
>
> [2] Gallego, Víctor, and David Rios Insua. “Stochastic Gradient MCMC with Repulsive Forces.” ArXiv: Machine Learning, 2018.
>
> [3] Liu, Qiang, and Dilin Wang. “Stein Variational Gradient Descent as Moment Matching.” Advances in Neural Information Processing Systems, vol. 31, 2018, pp. 8854–8863.

---

### Review · Ethics_Reviewer_Y5io · 2021-07-22

**Recommendation:** n/a

**Ethics Review:**

this was a misclick.
Only one reviewer requested ethics review, and they explicitly wrote that there were no issues.

---

### Review · Ethics_Reviewer_hoaL · 2021-08-13

**Recommendation:** n/a

**Ethics Review:**

No obvious ethical concerns.

---

### Decision · Program_Chairs · 2021-09-27

**Decision:**

Accept (Poster)

**Comment:**

This paper proposes an elegant generalisation of the Stein variational gradient descent method that introduces additional degrees of freedom into how the dynamics are defined, together with empirical evidence in support of this additional flexibility being practically useful. All reviewers agreed on the correctness of the method and that the paper is mostly well-written, but there was one reviewer who viewed the contribution as too incremental.  The paper addresses an important open question - deterministic alternatives to MCMC - which will be of general interest at NeurIPS.